# Navigating contradictions in enteric chemotactic stimuli

Kailie Franco[1], Zealon Gentry-Lear[2†], Michael Shavlik[3], Michael J Harms[2,3], Arden Baylink[1,4*]

[1]Department of Veterinary Microbiology and Pathology, Washington State University, Pullman, United States; [2]Institute of Molecular Biology, University of Oregon, Eugene, United States; [3]Department of Chemistry & Biochemistry, University of Oregon, Eugene, United States; [4]Amethyst Antimicrobials, LLC, Pullman, United States

## eLife Assessment

In this manuscript, Franco and colleagues present **compelling** evidence that fecal extracts containing high concentrations of indole, a known repellent, enhance rather than protect against invasion of colonic tissue by Salmonella. The authors describe **important** findings that lead to the conclusion that the competing effects of attractants present in fecal matter, including L-serine, also sensed by the Tsr chemoreceptor that senses indole, override the repulsive effect of indole.

***For correspondence:**
arden.baylink@wsu.edu

**Present address:** [†]Zealon Gentry-Lear, Department of Microbiology, University of Washington,Seattle, Washington, United States

**Abstract** Previously, we showed Enterobacteriaceae use chemotaxis and the chemoreceptor Tsr for attraction to blood serum (Glenn et al., 2024). Here, we investigated the complementary role of Tsr in mediating chemorepulsion, a behaviour by which motile bacteria avoid deleterious compounds to locate permissive niches. In the gut, indole is a bacteriostatic compound produced by microbiota and thought to act as a chemorepellent for invading pathogens, thereby protecting the host against infection. The principal reservoir of intestinal indole is fecal matter, a complex biological material that contains both attractant and repellent stimuli. Whether indole in its natural context is sufficient for pathogen chemorepulsion or host protection has remained unknown. Using an intestinal explant system, we show that pure indole suppresses an infection advantage mediated through chemotaxis for the enteric pathogen Salmonella Typhimurium, but this effect is abolished in the presence of other fecal chemoeffectors, including the chemoattractant L-Serine (L-Ser), dependent on the chemoreceptor Tsr. Live imaging reveals that although S. Typhimurium is repelled by pure indole, the pathogen is actually strongly attracted to human fecal matter despite its high indole content, and this response is mediated by Tsr, which simultaneously senses both stimuli. Fecal attraction is conserved across diverse Enterobacteriaceae species that harbor Tsr orthologues, including *Escherichia coli*, Citrobacter koseri, Enterobacter cloacae, and clinical isolates of non-typhoidal Salmonella. In a defined system of fecal chemoeffectors, L-Ser and other fecal chemoattractants override indole chemorepulsion, but the magnitude of bacterial chemoattraction is controlled by indole levels. Together, these findings suggest chemorepulsion elicited by indole is not protective against enteric infection and actually benefits pathogens by helping them locate niches with lower competitor density. Our study highlights the limitations of applying single-effector studies in predicting bacterial behavior in natural environments, where chemotaxis is shaped by the integration of multiple, often opposing, chemical signals.

## Introduction

Motile bacteria that colonize the gastrointestinal tracts of humans and other animals employ chemotaxis to sense chemical effectors in the gut lumen and navigate to environments conducive to growth and colonization (*Zhou et al., 2023*; *Keegstra et al., 2022*; *Bi and Sourjik, 2018*; *Matilla and Krell, 2018*; *Matilla et al., 2023*). This process is controlled by chemoreceptor proteins, which recognize chemical effectors and transduce signals through a phosphorylation cascade to regulate flagellar rotation and swimming direction, ultimately determining the spatial and temporal patterns of bacterial colonization (*Figure 1A*; *Zhou et al., 2023*; *Keegstra et al., 2022*; *Matilla and Krell, 2017*; *Ortega et al., 2017*). While many effectors have been studied and characterized in isolation as chemoattractants or chemorepellents (*Matilla and Krell, 2018*; *Ortega et al., 2017*), natural environments like the gut contain complex mixtures of opposing signals. Only a handful of studies have investigated how bacteria navigate conflicting chemical gradients, and much remains to be learned about how bacteria prioritize these signals to direct their movement and colonization (*Figure 1A*; *Zhou et al., 2023*; *Matilla and Krell, 2018*; *Yang et al., 2020*; *Livne and Vaknin, 2022*; *Keller and Segel, 1971*; *Adler and Tso, 1974*; *Zhang et al., 2019*; *Kalinin et al., 2010*; *Englert et al., 2009*; *Huang et al., 2017*).

A chemical effector of major importance for enteric bacterial communities is indole, an interbacterial signaling molecule that regulates diverse aspects of physiology and lifestyle (*Li et al., 2021*; *Lee and Lee, 2010*; *Gupta et al., 2022*). Indole is excreted by gut microbiota as a byproduct of tryptophan metabolism and accumulates to millimolar levels in human feces (*Figure 1A*; *Li et al., 2021*; *Darkoh et al., 2015*; *Chappell et al., 2016*). Indole is amphipathic and can transit bacterial membranes to regulate biofilm formation and motility, suppress virulence programs, and exert bacteriostatic and bactericidal effects at high concentrations (*Li et al., 2021*; *Lee and Lee, 2010*; *Gupta et al., 2022*; *Chappell et al., 2016*; *Ye et al., 2022*; *Kumar and Sperandio, 2019*; *Kohli et al., 2018*; *Nikaido et al., 2012*). Indole was one of the earliest identified chemorepellents, and subsequent work has extensively explored its role in *Escherichia coli* chemotaxis, mostly examining responses to indole as a singular effector (*Supplementary file 1*; *Yang et al., 2020*; *Livne and Vaknin, 2022*; *Adler and Tso, 1974*; *Englert et al., 2009*; *Gupta et al., 2022*; *Bansal et al., 2007*). Recent studies have advanced understanding of the molecular mechanisms underlying *E. coli* indole taxis and the involvement of the chemoreceptor taxis to serine and repellents (Tsr) (*Figure 1A*; *Yang et al., 2020*; *Livne and Vaknin, 2022*; *Englert et al., 2009*; *Gupta et al., 2022*).

From this body of research, the hypothesis emerged that indole from the gut microbiota functions to repel pathogens and restrict their growth as a mechanism of colonization resistance (*Yang et al., 2020*; *Li et al., 2021*; *Lee and Lee, 2010*; *Gupta et al., 2022*; *Kohli et al., 2018*; *Nikaido et al., 2012*). If true, this could represent an intriguing avenue for cultivating microbiomes that are more robust against pathogen infiltration—a major area of interest for improving gut health (*Bai et al., 2023*; *Wolter et al., 2021*). However, this hypothesis is based largely on observations of bacterial chemorepulsion in response to indole as a single effector, and it remains unclear whether chemotactic behavior is similar or altered in the presence of other intestinal effectors. For instance, fecal material, while rich in indole, also contains high concentrations of sugars and amino acids that serve as nutrients and chemoattractants—factors that could diminish, nullify, or have no influence on indole chemorepulsion (*Zhou et al., 2023*; *Ye et al., 2022*; *Kumar and Sperandio, 2019*; *Wishart et al., 2022*). Although indole may suppress bacterial growth, nutrients derived from the host diet could offset this effect, allowing bacteria to tolerate indole and still benefit from colonizing indole-rich niches. Indeed, pathogens frequently succeed in establishing enteric infections, suggesting that they can tolerate indole or circumvent its effects under certain conditions. Thus, bacteria in the intestinal environment must navigate contradictory chemotaxis signals, and how they resolve these conflicts influences infection, pathogenesis, and host health in ways that remain to be elucidated. Furthermore, because indole taxis has only been studied in *E. coli*, it remains unconfirmed whether other enteric species even chemotactically sense or respond to indole (*Supplementary file 1*).

In this study, we aimed to (1) test the hypothesis that indole taxis protects against intestinal infection and (2) determine how enteric pathogens use chemotaxis to navigate the complex mixture of opposing chemical cues present in fecal material, a major source of both indole and nutrients within the gut. We used *Salmonella enterica* serovar Typhimurium as a model pathogen, as it requires chemotaxis and the chemoreceptor Tsr for efficient infection and cellular invasion of intestinal tissue (*Rogers et al., 2021*; *Rivera-Chávez et al., 2013*; *Rivera-Chávez et al., 2016*; *Gül et al., 2024*;

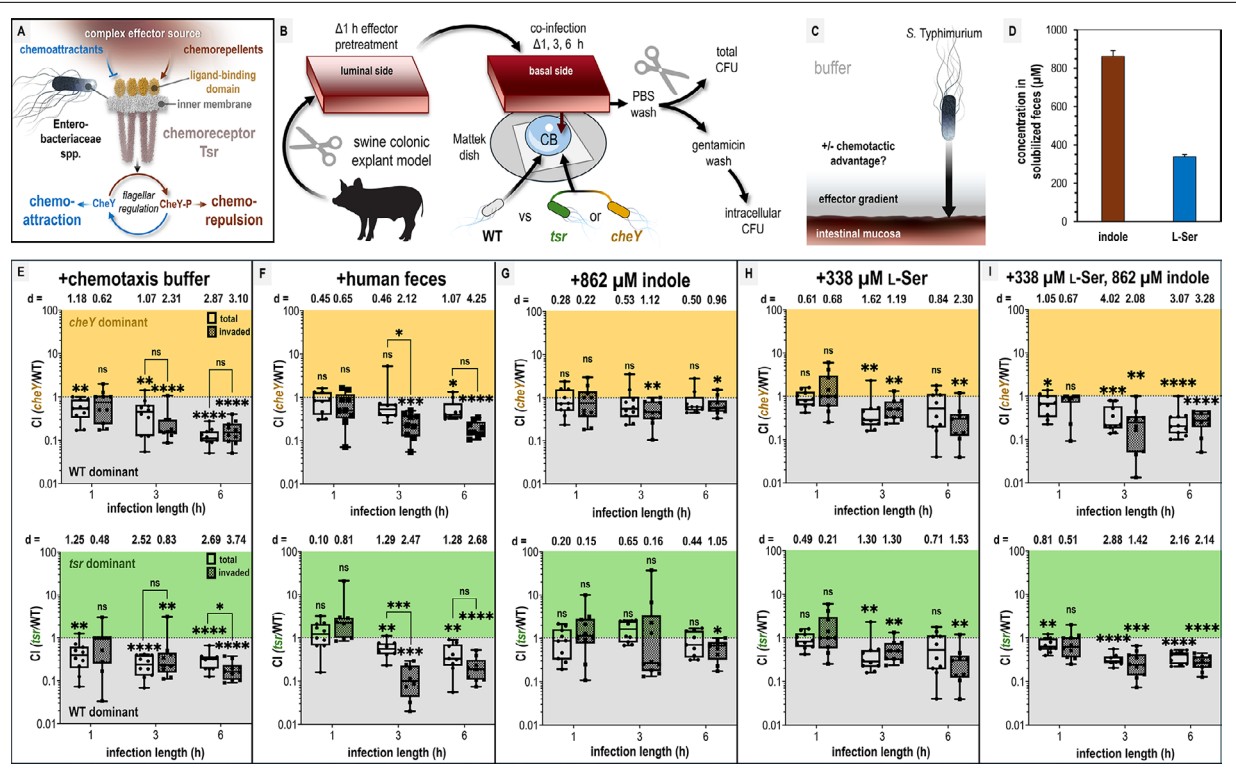

**Figure 1.** Chemotaxis-mediated infection advantages in the presence of fecal effectors. (**A**) Overview of the role of Tsr in coordinating responses to conflicting stimuli. (**B**) Experimental design of colonic explant infections. See Materials and methods for experimental details such as tissue dimensions. (**C**) Conceptual model of the explant infection system. The effectors from the treated tissue (gray) diffuse into the surrounding buffer solution providing a gradient. Note that the bacteria are not immersed in the effector solution and experience a local concentration during infection much lower than the effector pretreatment. Quantifications of tissue-associated bacteria reflect the ability of chemotaxis to provide an advantage (black arrow) in accessing the intestinal mucosa (reddish brown). (**D**) Serine (presumed to be nearly 100% L-Ser, see Materials and methods) and indole content of liquid human fecal treatments, as measured by mass spectrometry. (**E–I**) Competitive indices (CIs) of colony-forming units (CFUs) recovered from co-infected swine explant tissue, either from the total homogenate (open box and whiskers plots), or from tissue washed with gentamicin to kill extracellular and attached cells, which we refer to as the 'invaded' intracellular population (checkered box and whisker plots), as indicated. Each data point represents a single experiment of a section of tissue infected with bacteria, normalized by tissue weight, and the CI of CFUs recovered from that tissue ($n = 7$–$10$). Boxes show median values (line) and upper and lower quartiles, and whiskers show max and min values. Effect size (Cohen's $d$) and statistical significance are noted for each experiment in relation to competitive advantage, that is deviation from a CI of 1 (not significant, ns; *$p < 0.05$, **$p < 0.01$, ***$p < 0.001$, ****$p < 0.0001$). See also *Figure 1 - figure supplement 1* for competition between wildtype (WT) and an invasion-inhibited mutant *invA*, and *Figure 1 - figure supplement 2* for disaggregated CFU enumerations for each experimental group prior to CI calculation. *Source data 1* contains all numerical CFU measurements.

The online version of this article includes the following figure supplement(s) for figure 1:

**Figure supplement 1.** Competition of wildtype (WT) and *invA* in explant infections.

**Figure supplement 2.** Colony-forming units (CFUs) recovered from swine colonic explant infections.

*Winter et al., 2013*; *Winter et al., 2010*; *Cooper et al., 2021*). Tsr is of particular interest because it mediates responses to both chemorepellents and the chemoattractant L-Serine (L-Ser), suggesting an important role in integrating contradictory chemotactic signals for *S.* Typhimurium and other *Enterobacteriaceae* that possess Tsr orthologs (*Figure 1A, B*; *Glenn et al., 2024*; *Zhou et al., 2023*; *Yang et al., 2020*; *Livne and Vaknin, 2022*; *Englert et al., 2009*; *Gupta et al., 2022*; *Bansal et al., 2007*; *Piñas et al., 2022*; *Burt et al., 2020*; *Chen et al., 2022*). *S.* Typhimurium also differs from *E. coli* in that it lacks tryptophanase and cannot itself produce indole, thereby offering a novel perspective on indole taxis (*Vega et al., 2013*; *Roager and Licht, 2018*). Our findings reveal that bacterial chemotaxis within biologically relevant mixtures of effectors cannot be reliably inferred from studies of individual compounds alone, with important implications for understanding how chemotaxis influences pathogen behavior within the gut.

## Results

### Fecal indole is insufficient to protect against pathogen invasion in a colonic explant model

We sought to test whether indole in human fecal matter protects against *S. enterica* serovar Typhimurium infection and whether this involves indole chemorepulsion mediated by the chemoreceptor Tsr. *Salmonella* Typhimurium preferentially infects tissue of the distal ileum but also infects the cecum and colon in humans and animal models (*Chowdhury et al., 2023*; *Ehrhardt and Grassl, 2022*; *Grassl et al., 2008*; *Radlinski et al., 2024*; *Mandal and Mani, 1976*). We presume that the amount of indole is greatest in the lower gastrointestinal tract, where tryptophan has been maximally digested by microbial tryptophanases. To mimic this environment, we developed a swine colonic explant model that simulates the architecture and size of adult human colonic tissue (*Figure 1B*; *Roager and Licht, 2018*; *Kronsteiner et al., 2013*; *Smith and Swindle, 2006*; *Poutahidis et al., 2001*; *Boyen et al., 2009*; *Boyen et al., 2008*). This model was based on a prior study using explant tissue to characterize cellular invasion via gentamicin washes, which kill extracellular and surface-attached bacteria (*Figure 1B*; *Köppen et al., 2023*). Gentamicin washing is also a commonly used method to quantify intracellular *Salmonella* Typhimurium populations in cell culture experiments (*Bomjan et al., 2019*; *Birhanu et al., 2018*).

A section of colonic tissue was prepared for each experiment by gentle cleaning and then soaked with different effector solutions for 1 hr: solubilized human feces, purified indole and/or L-Ser at fecal-relevant concentrations, or buffer as a control (see Materials and methods for tissue dimensions and additional experimental details). Subsequently, the tissue was removed from the effector treatment and oriented with the luminal side downward onto a Mattek dish containing 300 µl of buffer with motile *S.* Typhimurium (*Figure 1B*). To be clear, in this system, the bacteria are not immersed in the effector treatment and experience an effector concentration far lower than used for the soak of the tissue prior to infection where the residual effector diffuses outward from the tissue into the much larger volume of buffer in which the cells are swimming (*Figure 1C*). This establishes a chemical gradient which we can use to quantify the degree to which different effector treatments are permissive of pathogen association with, and cellular invasion of, the intestinal mucosa (*Figure 1C*). Using this approach, we sought to test infection based on fecal treatments and fecal-relevant concentrations of L-Ser and indole (*Figure 1D*).

We employed a strategy of co-infections in order to compete and compare the advantages conferred by chemotaxis using *S.* Typhimurium strain IR715 wildtype (WT) and either a *cheY* mutant (motile but non-responsive to chemoeffector stimuli) or *tsr* deletion mutant (*Figure 1B*, see Key Resources Table, *Rivera-Chávez et al., 2013*). The functionality of these mutants has been previously confirmed through in vivo infection studies using genetically complemented strains (*Rivera-Chávez et al., 2013*; *Rivera-Chávez et al., 2016*). To assess the role of chemotaxis in infection, we quantified bacteria harvested from tissue homogenates at 1, 3, and 6 hr post-infection (*Figure 1C*, Materials and methods, *MacBeth and Lee, 1993*; *Sharma and Puhar, 2019*). Using this experimental setup, we found for tissue pretreated with fecal material that WT had a competitive advantage over an invasion-inhibited mutant (*invA*) in homogenates from gentamicin-washed tissue, but no advantage in unwashed homogenates, supporting that gentamicin washing selects for intracellular bacteria (*Thiennimitr et al., 2011*). For simplicity in discussing the explant infection data, we refer to these two types of quantifications as 'invaded' (i.e. *Salmonella* that have entered non-phagocytic host cells) and 'total' bacteria in the figures, respectively (*Figure 1—figure supplement 1*; *Haneda et al., 2009*).

In buffer-treated explant experiments, WT *S.* Typhimurium exhibits a 5- to 10-fold time-dependent advantage in colonization and cellular invasion compared to chemotactic mutants, indicating that chemotaxis, and specifically Tsr, promotes tissue colonization in this system (*Figure 1E*, *Figure 1—figure supplement 2A, B*). The mechanism mediating this advantage is not clear, but could arise from a combination of factors, including sensing of effectors emitted from the tissue, redox or energy taxis, and/or swimming behaviors that enhance infection (*Matilla et al., 2023*; *Rivera-Chávez et al., 2013*; *Rivera-Chávez et al., 2016*; *Cooper et al., 2021*). This experiment indicates that under baseline conditions, the intestinal mucosa is accessible to the pathogen. The hypothesis that indole protects against pathogen colonization predicts that feces, the major biological reservoir of gut indole, should confer protection against infection. Contrary to this prediction, we found that fecal treatment provided a similar infection advantage as buffer treatment, and this effect was mediated by chemotaxis and Tsr

(*Figure 1F*, *Figure 1—figure supplement 2C, D*). One notable difference, however, was that fecal treatments yielded a higher competitive advantage for the WT invaded population over the total population at 3 hr compared to buffer treatment (comparing buffer and fecal treatments: WT vs *cheY*, p = 0.18 and p = 0.02; WT vs *tsr*, p = 0.35 and p = 0.0004, respectively; *Figure 1E, F*). That the phenotype fades at longer time points could relate to the effector gradient being eliminated by diffusion.

Analysis of the liquefied human fecal matter used in this study revealed an indole concentration of 862 µM, consistent with previously reported ranges (0.5–5 mM) (*Figure 1D*; Materials and methods) (*Li et al., 2021*; *Darkoh et al., 2015*; *Chappell et al., 2016*; *Ye et al., 2022*; *Kumar and Sperandio, 2019*). When colonic tissue was treated with purified indole at this concentration, WT loses its competitive advantage over the chemotactic mutants (*Figure 1G*, *Figure 1—figure supplement 2E, F*). Given that Tsr mediates attraction to L-Ser in both *E. coli* and *Salmonella* Typhimurium, we hypothesized that L-Ser present in feces might negate the protective effect of indole (*Glenn et al., 2024*; *Zhou et al., 2023*; *Kitamoto et al., 2020*; *Sugihara and Kamada, 2020*). Treatment with 338 µM L-Ser alone, the concentration present in our fecal sample (*Figure 1D*; Materials and methods), conferred a WT advantage similar to buffer and fecal treatments (*Figure 1H*) and WT also exhibits a colonization advantage when L-Ser is co-administered with indole (*Figure 1I*).

Our key takeaway from these experiments is that pretreatment of intestinal tissue with indole alone is unique in that the WT strain gains no infection advantage in this context (*Figure 1G*). In contrast, under all other treatment conditions, the WT infects the tissue to a greater extent than the chemotactic mutants (*Figure 1E–I*). Put another way, chemotaxis and Tsr enhance pathogen transit of the chemical gradient and increase access to the intestinal tissue in all conditions except when indole is the sole effector. This was surprising, given that other treatments contain the same concentration of indole and still permit a chemotaxis- and Tsr-mediated advantage (*Figure 1*). To us, this suggests that bacterial perception of indole via chemotaxis differs fundamentally depending on whether indole is the only effector present or amidst other fecal effectors. Notably, only fecal treatment resulted in a greater competitive advantage for the invaded population than for the total population (*Figure 1F*).

## Enterobacteriaceae species are attracted to human feces despite high indole content

Having found that chemotaxis and Tsr mediate efficient infection, but do not confer an advantage when indole is the only effector, we next sought to understand the chemotactic behaviors orchestrated by Tsr in response to indole-rich feces. Given that feces represents the highest concentration of indole that *S.* Typhimurium is likely to encounter natively, we expected to observe chemorepulsion (*Figure 1D*, ; *Yang et al., 2020*; *Livne and Vaknin, 2022*; *Englert et al., 2009*; *Bansal et al., 2007*; *Chen et al., 2022*). We employed the chemosensory injection rig assay (CIRA) for live imaging of bacterial chemotaxis responses to a source of effectors injected through a glass microcapillary (*Glenn et al., 2024*). The flow and dynamic nature of the gut lumen make this a suitable in vitro approach for modeling and studying enteric chemotaxis responses (*Glenn et al., 2024*; *Perkins et al., 2019*).

In this assay, chemoattraction is observed as an influx of cells toward the effector source, whereas chemorepulsion is indicated by a decrease in local cell density (*Figure 2—figure supplement 1*). As described previously, effector injection creates a steep chemical microgradient (*Glenn et al., 2024*). Using mathematical modeling of the diffusion of fecal-relevant concentrations of indole and L-Ser, we approximated the local concentrations experienced by bacteria at varying distances from the injection site. For most of the field of view, these concentrations fall within the picomolar to low nanomolar range (*Figure 2A*; Materials and methods, *Glenn et al., 2024*).

Over 5 min, we found both WT and *tsr* exhibit strong chemoattraction to fecal material, whereas *cheY* remains randomly distributed (*Figure 2B*, *Video 1*). By examining the radial distribution of the bacterial populations, we found WT more tightly centers around the treatment source than *tsr* (*Figure 2C–E*, *Video 1*). In terms of the rate of bacterial accumulation, the chemoattraction of *tsr* lags behind the WT for the first 120 s (*Figure 2F, G*, *Video 1*). We wondered how these deficiencies in fecal attraction might translate to direct competition, where different strains are experiencing the same treatment source simultaneously. To address this, we performed CIRA with solubilized human feces and two strains present in the same pond, which we tracked independently through fluorescent markers (*Figure 3*; *Glenn et al., 2024*). As expected, WT shows a strong chemoattraction response versus *cheY* (*Figure 3A*, *Video 2*). Interestingly, we found that when competed directly, WT vastly

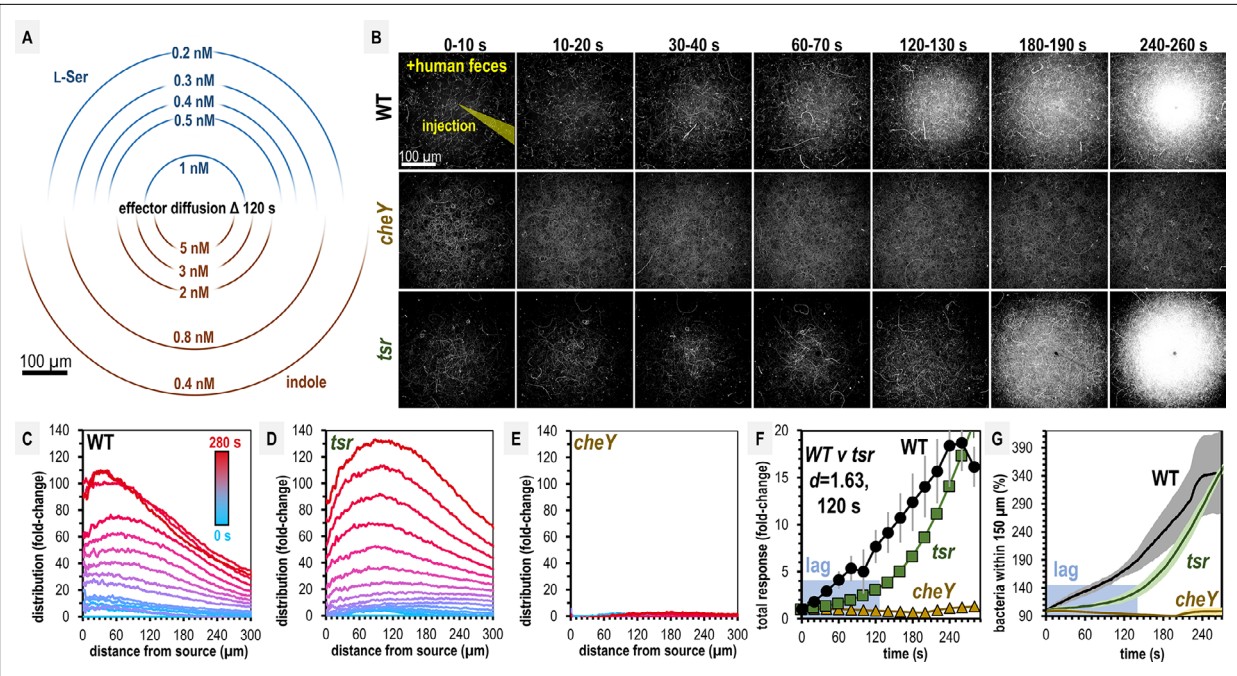

**Figure 2.** *Salmonella* Typhimurium exhibits attraction toward indole-rich liquid human fecal material. (**A**) Diffusion modeling showing calculated local concentrations in chemosensory injection rig assay (CIRA) experiments with liquid human fecal material based on distance from the central injection source. (**B**) Max projections of representative *S.* Typhimurium IR715 responses to a central source of injected liquid human fecal material. (**C**–**E**) Bacterial population density over time in response to fecal treatment. The initial uniform population density in these plots is indicated with the blue line (time 0), and the final mean distributions with the red line (time 280 s), with the mean distributions between these displayed as a blue-to-red spectrum at 10-s intervals. (**F**–**G**) Temporal analyses of area under the curve (AUC) or relative number of bacteria within 150 µm of the source. Effect size (Cohen's *d*) comparing responses of wildtype (WT) and *tsr* attraction at 120 s post-treatment is indicated. Data were collected at 30°C. Data are means and error bars are standard error of the mean (SEM, n = 3–5). See also *Figure 2—figure supplement 1*, *Video 1*, *Supplementary file 1*.

The online version of this article includes the following figure supplement(s) for figure 2:

**Figure supplement 1.** Chemosensory injection rig assay (CIRA) design, diffusion modeling, and responses to L-Ser or indole.

---

outperforms *tsr*, with the maximal bacterial distribution in proximity to the treatment source higher by about fourfold (*Figure 3B*, *Video 2*). These data confirm that despite its high indole content, *S.* Typhimurium is attracted to human fecal material through chemotaxis, and this response involves Tsr, although not as the sole mediator. We expect the attraction of the *tsr* mutant is explained by the fact that *S.* Typhimurium possesses other chemoreceptors that detect glucose, galactose, ribose, and L-Asp as chemoattractants, which are also present in human feces (*Zhou et al., 2023*; *Ortega et al., 2017*; *Ranque et al., 2020*; *Ke et al., 2024*; *Larke et al., 2023*; *Shen et al., 2022*; *Yoo et al., 2024*).

Recent work highlights how genetic diversity among *Salmonella* strains and differences in Tsr expression, even within isogenic populations, modulate chemotaxis function (*Livne and Vaknin, 2022*; *Gül et al., 2024*). To gain a broader perspective on fecal taxis, we examined the responses among diverse non-typhoidal *Salmonella* serovars and strains responsible for human infections. Using dual-channel imaging, we compared *S.* Typhimurium IR715 with a clinical isolate of *S.* Typhimurium, SARA1, and found both strains exhibit attraction to feces, although SARA1 shows a slightly weaker response (*Figure 3C*,

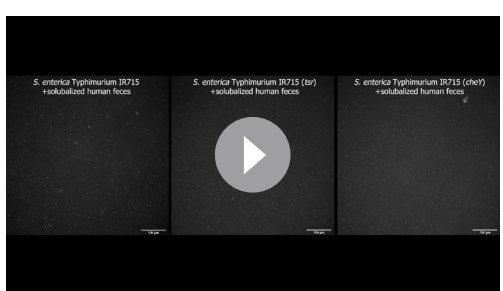

**Video 1.** Chemotactic response of *S.* Typhimurium IR715 to solubilized human feces. Representative chemosensory injection rig assay (CIRA) experiments showing *S.* Typhimurium IR715 wildtype (WT) and mutant strains responding to a source over 300 s (shown at 10× speed). Viewable at: https://www.youtube.com/watch?v=BqUcRN3YwjU.
https://elifesciences.org/articles/106261/figures#video1

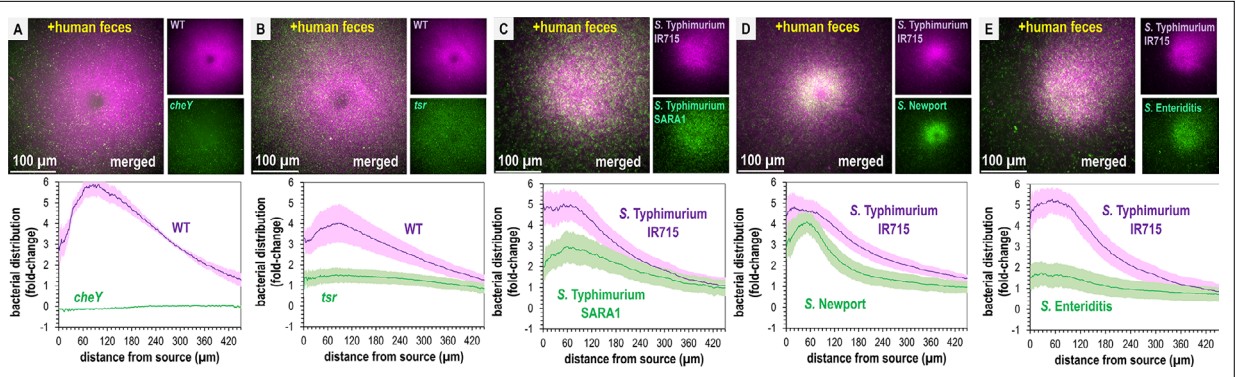

**Figure 3.** Non-typhoidal *Salmonella* exhibit fecal attraction. (**A–E**) Dual-channel imaging of chemotactic responses to solubilized human feces by wildtype (WT) *S.* Typhimurium IR715 (pink) and isogenic mutants or clinical isolate strains (green), as indicated. Max projections are shown at time 295–300 s post-treatment. Data were collected at 37°C. Data are means and error bars are standard error of the mean (SEM, *n* = 3–5). See also *Videos 2 and 3*.

*Video 3*, *Beltran et al., 1991*). We then tested a clinical isolate of *S.* Newport, an emerging cause of salmonellosis in the United States and Europe (*Shariat et al., 2013*; *Ferrari et al., 2019*). This strain is strongly attracted to fecal material, with a tighter accumulation of cells at the treatment source than *S.* Typhimurium IR715 (*Figure 3D*, *Video 3*). We also examined a clinical isolate of *S.* Enteritidis, a zoonotic pathogen commonly transmitted from poultry, which displays weak attraction to fecal material (*Figure 3E*, *Video 3*, *Ferrari et al., 2019*).

Next, we extended this analysis to other disease-causing *Enterobacteriaceae* that possess Tsr orthologs (*Glenn et al., 2024*). *E. coli* strain MG1655, commonly used for in vitro experiments, and *E. coli* NCTC 9001, a strain isolated from human urine and associated with urinary tract infections, both exhibited fecal attraction, although the response was more diffuse than that observed for *Salmonella* (*Figure 4A–D*; *Videos 4 and 5*). The clinical isolate *Citrobacter koseri* strain 4225-83 also showed fecal attraction, with a tight association near the effector source (*Figure 4E, F*; *Video 6*). Lastly, *Enterobacter cloacae* CDC 442-68, a clinical isolate with uncharacterized chemotaxis behavior, appeared to exhibit fecal attraction as well, although this strain was not extensively tested due to limited motility under laboratory conditions (*Figure 4—figure supplement 1*).

Overall, we find that Tsr mediates fecal attraction in *Salmonella*, and that this behavior is conserved among diverse *Enterobacteriaceae* that possess Tsr and are associated with human infections.

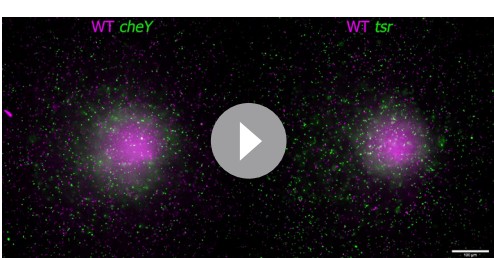

**Video 2.** Chemotactic response of *S.* Typhimurium IR715 wildtype (WT) and chemotactic mutant strains to solubilized human feces. Representative chemosensory injection rig assay (CIRA) experiments showing competition between *S.* Typhimurium IR715 (mPlum) and *cheY*, or *tsr*, as indicated (GFP), over 300 s. Viewable at: https://www.youtube.com/watch?v=D5JL46b4lsI.

https://elifesciences.org/articles/106261/figures#video2

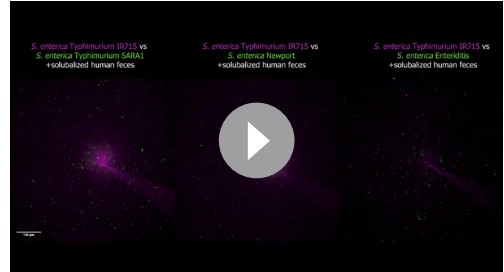

**Video 3.** Chemotactic response of *S. enterica* clinical isolates to solubilized human feces. Representative chemosensory injection rig assay (CIRA) experiments showing competition between *S.* Typhimurium IR715 (mPlum) and clinical isolates, as indicated (GFP), responding to a source of solubilized human feces over 300 s. Viewable at: https://www.youtube.com/watch?v=dLsFDV0XgpY.

https://elifesciences.org/articles/106261/figures#video3

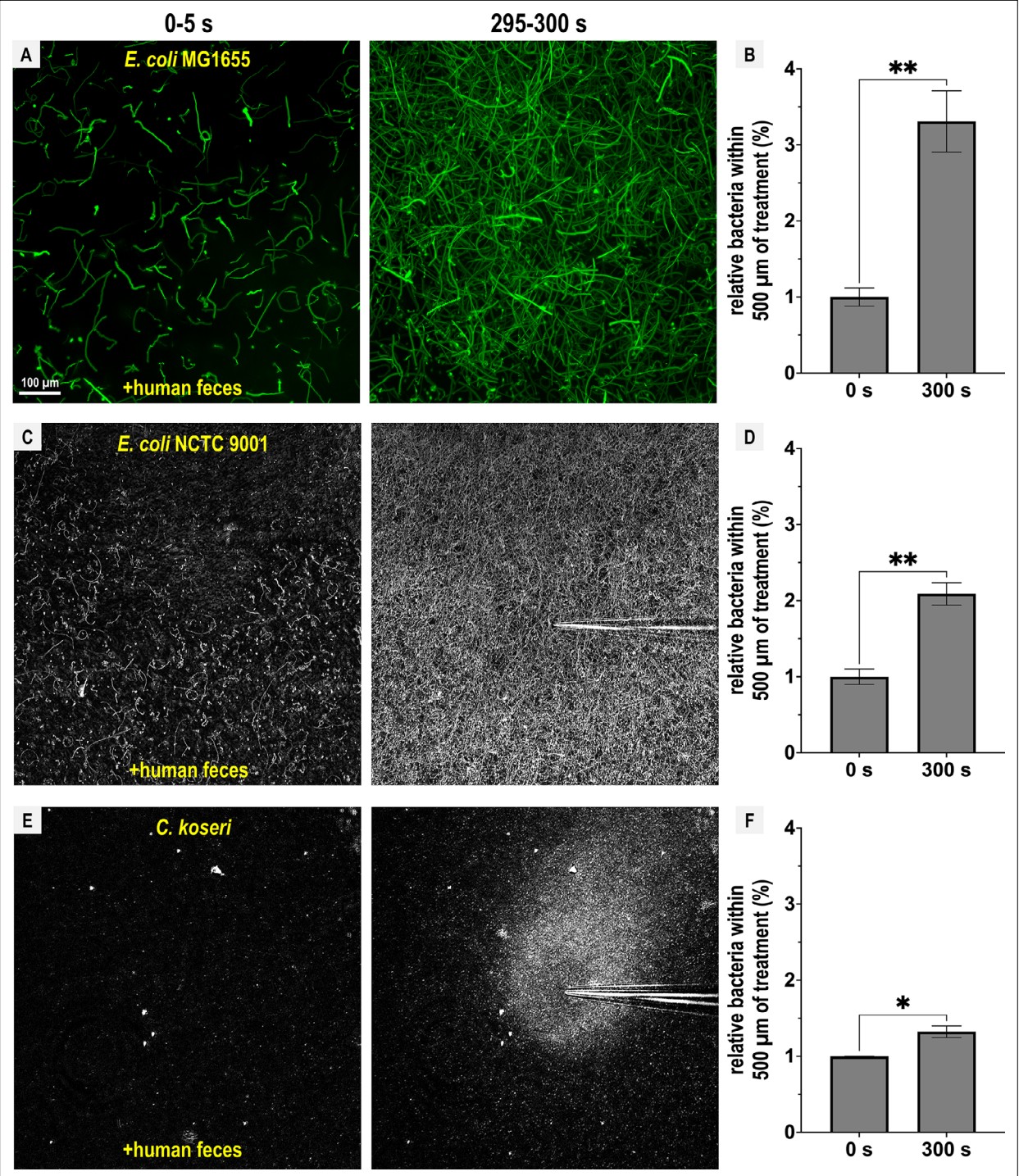

**Figure 4.** Representative *Enterobacteriaceae* exhibit fecal attraction. Max projections are shown from chemosensory injection rig assay (CIRA) experiments over 5 s before fecal treatment and after 5 min of treatment, as well as quantifications of bacteria within 500 µm of the treatment source at these same time points for *E. coli* MG1655 (**A**, **B**, GFP-reporter), *E. coli* NCTC 9001 (**C**, **D**, phase), and *C. koseri* CDC 4225-83 (**E**, **F**, phase). Data were collected at 37°C. Data are means and error bars are standard error of the mean (SEM, *n* = 3–5) with statistical significances denoted (not significant, ns; *p < 0.05, **p < 0.01, ***p < 0.001, ****p < 0.0001). See also *Videos 4–6*.

The online version of this article includes the following figure supplement(s) for figure 4:

**Figure supplement 1.** *Enterobacter cloacae* exhibits fecal attraction.

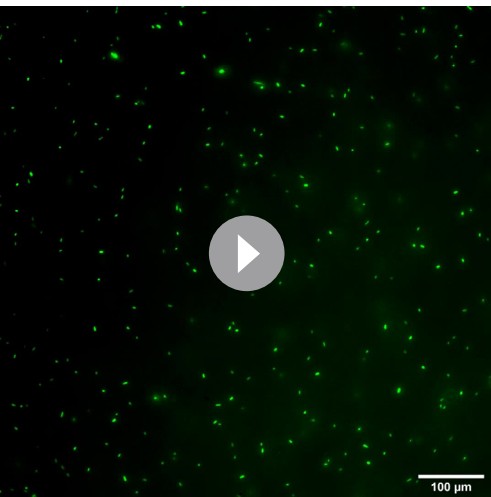

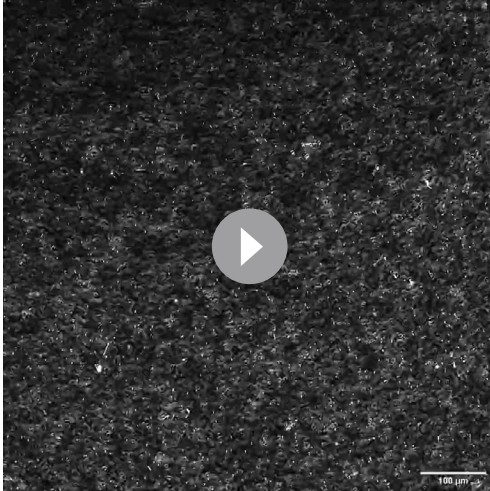

**Video 4.** Chemotactic response of *E. coli* MG1655 to solubilized human feces. Representative chemosensory injection rig assay (CIRA) experiment over 300 s. Viewable at: https://youtube.com/shorts/WH6tabDbrw4?feature=share.

https://elifesciences.org/articles/106261/figures#video4

**Video 5.** Chemotactic response of *E. coli* NCTC 9001 to solubilized human feces. Representative chemosensory injection rig assay (CIRA) experiment over 300 s. Viewable at: https://youtube.com/shorts/yzU2M4Z_Yf4?feature=share.

https://elifesciences.org/articles/106261/figures#video5

Although the degree of attraction varies, none of the enteric pathogens or pathobionts tested exhibited chemorepulsion from feces, despite its high indole content.

## Fecal chemoattractants override indole chemorepulsion

To better understand the relationship between indole and other fecal effectors in directing *S.* Typhimurium chemotaxis, we next employed a reductionist approach and developed a mixture of fecal effectors at physiological concentrations based on our measurements and the Human Metabolome Database (*Figure 5*, *Wishart et al., 2022*). Along with the chemorepellent indole (862 µM), we tested combinations of fecal chemoattractants including L-Ser (338 µM), sensed through Tsr; D-glucose (970 µM), D-galactose (78 µM), and ribose (28.6 µM), sensed through the chemoreceptor Trg; and L-aspartate (L-Asp, 13 µM), sensed through the chemoreceptor Tar (*Glenn et al., 2024*; *Zhou et al., 2023*; *Ortega et al., 2017*; *Wishart et al., 2022*; *Falke and Hazelbauer, 2001*; *Lai et al., 2005*). A low density of motile cells ($A_{600} \sim 0.05$) was used in the CIRA experiments to increase sensitivity for detecting attraction in response to different combinations of these fecal effectors (*Figure 5*).

We observed that L-Ser was sufficient to negate indole chemorepulsion and may even elicit attraction, although this was not statistically significant (*Figure 5A, B*; *Video 7*). When all effectors were present, bacteria were clearly attracted to the treatment (*Figure 5C, D*; *Video 8*), with a slightly reduced attraction in the absence of L-Ser (*Figure 5E, F*; *Video 9*). Interestingly, when all effectors were present but the concentration of indole was halved (431 µM), cells exhibited the greatest degree of attraction (*Figure 5G, H*; *Video 10*).

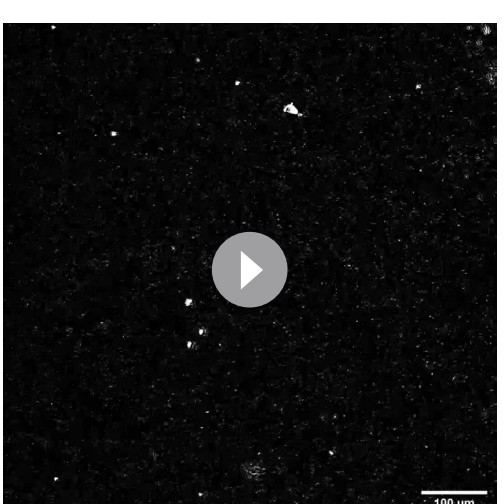

**Video 6.** Chemotactic response of *C. koseri* 4225-83 to solubilized human feces. Representative chemosensory injection rig assay (CIRA) experiments with treatment sources as indicated, over 300 s. Viewable at: https://youtube.com/shorts/s_ybO0xcIDw?feature=share.

https://elifesciences.org/articles/106261/figures#video6

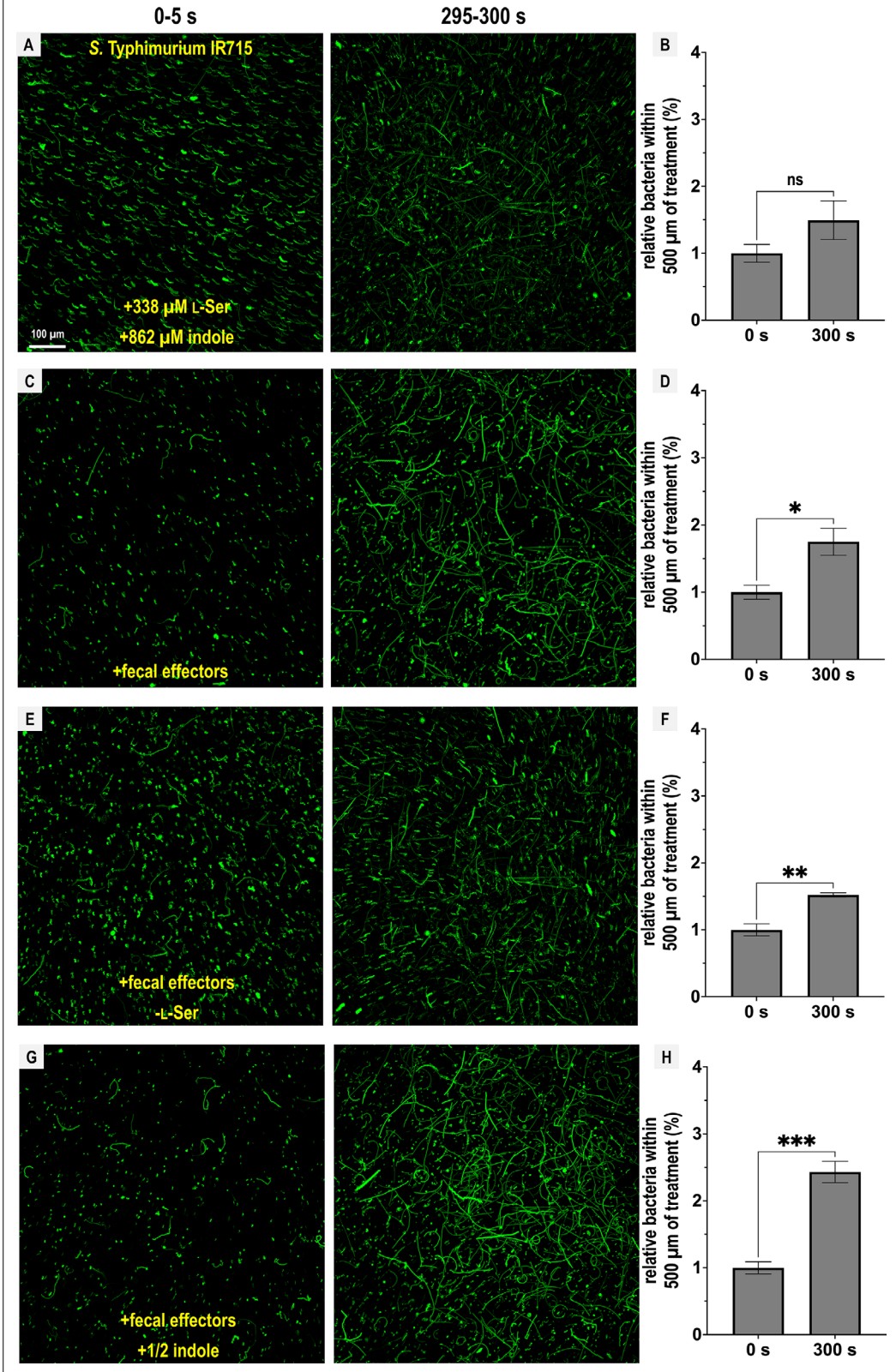

**Figure 5.** Chemotactic responses to defined fecal effector mixtures. Chemosensory injection rig assay (CIRA) experiments with *S.* Typhimurium IR715 were performed with different combinations of fecal effectors (*n* = 3–5). Max projections are shown from experiments over 5 s before fecal treatment and after 5 min of treatment as well as quantifications of bacteria within 500 μm of the treatment source at these same time points. Data are means

*Figure 5 continued on next page*

*Figure 5 continued*

and error bars are standard error of the mean (SEM, *n* = 3–5), with statistical significance denoted (not significant, ns; *p < 0.05, **p < 0.01, ***p < 0.001, ****p < 0.0001). To achieve the greatest degree of sensitivity to differences in responses, experiments were performed using the same culture on the same day. The complete fecal effector mixture consists of indole (862 µM), L-Ser (338 µM), D-glucose (970 µM), D-galactose (78 µM), ribose (28.6 µM), and L-Asp (13 µM), modified to include or exclude certain effectors as indicated. See also *Videos 7–10*. Data were collected at 30°C.

From these data, we conclude that the Tsr ligand L-Ser can override chemorepulsion from indole. However, this effect can also be mediated by other fecal effectors sensed through different chemoreceptors, providing an explanation for the reduced, but still appreciable, fecal attraction observed for the *tsr* mutant (*Figure 3B*). While the overall responses to this mixture of fecal effectors can be characterized as attraction, the bacteria remain sensitive to indole levels, as reflected in the enhanced attraction observed in treatments with lower indole concentrations (*Figure 5G, H*).

## Mediation of opposing chemotactic responses by Tsr

We considered whether our inability to observe repulsion from fecal material and mixtures of fecal effectors might be due to *S.* Typhimurium not sensing indole as a chemorepellent, since this chemotactic response has only been previously described for *E. coli* (*Supplementary file 1*). We compared chemotaxis responses to either 5 mM L-Ser or 5 mM indole and found that *S.* Typhimurium responds rapidly to these two effectors as chemoattractants and chemorepellents, respectively (*Figure 2 -figure supplement 1H, I*). Treatment with 5 mM indole, a concentration at the upper end of what occurs in the human gut (*Kumar and Sperandio, 2019*), induces rapid chemorepulsion with the bacteria vacating the region proximal to the source (*Figure 2—figure supplement 1I*). Interestingly, the chemorepulsion response occurs faster than chemoattraction, with a zone of avoidance clearly visible within the first 10 s of indole exposure (*Figure 2—figure supplement 1H, I*, *Video 11*).

We next wondered if perhaps our fecal treatments contained insufficient indole to elicit chemorepulsion from *S.* Typhimurium. To identify the effective source concentrations that drive indole chemorepulsion and understand the temporal dynamics of this response, we performed a titration of indole across 0.05–10 mM (*Figure 6A, B*). At all concentrations tested, indole induces chemorepulsion, and the bacteria avoid the treatment source for the duration of the 5-min experiment (*Figure 6A,*

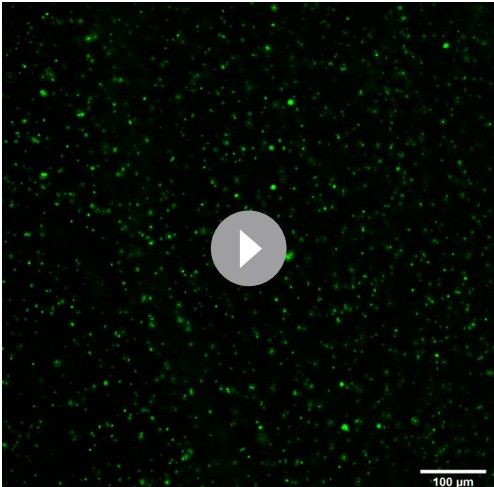

**Video 7.** Chemotactic response of *S.* Typhimurium IR715 to L-Ser and indole treatment at fecal-relevant concentrations. Representative chemosensory injection rig assay (CIRA) experiment over 300 s. Viewable at: https://youtube.com/shorts/4UEYoBS6jIQ?feature=share.

https://elifesciences.org/articles/106261/figures#video7

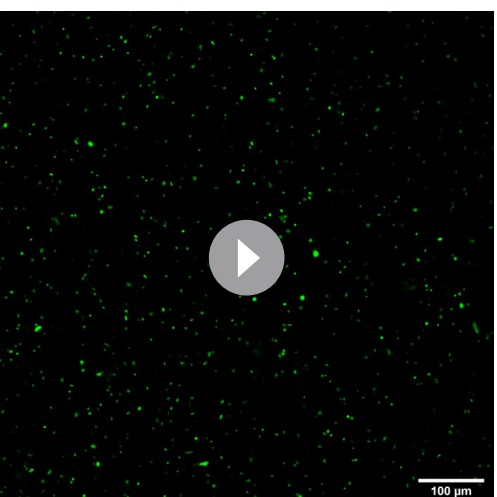

**Video 8.** Chemotactic response of *S.* Typhimurium IR715 to complete mixture of fecal effectors. Representative chemosensory injection rig assay (CIRA) experiment over 300 s. Viewable at: https://youtube.com/shorts/Yd14m3sI6Pw?feature=share.

https://elifesciences.org/articles/106261/figures#video8

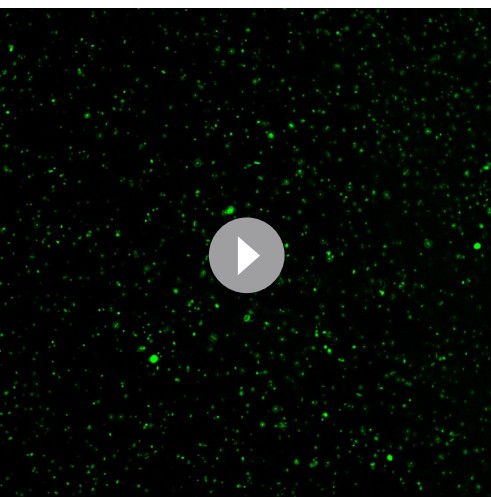

**Video 9.** Chemotactic response of *S.* Typhimurium IR715 to a mixture of fecal effectors lacking L-Ser. Representative chemosensory injection rig assay (CIRA) experiment over 300 s. Viewable at: https://youtu.be/5QM116BrHhQ.

https://elifesciences.org/articles/106261/figures#video9

*B*). At source concentrations exceeding 3 mM most motile cells vacate the field of view within 60 s (*Figure 6A*). Integrating these chemorepulsion responses and fitting them to a Monod curve suggests that an indole source concentration of approximately 67 µM is sufficient for half-maximal ($K_{1/2}$) chemorepulsion (*Figure 6C*). These data show that even though we observe strong attraction to fecal material, pure indole at the concentration present in fecal material, and far lower, is indeed a strong chemorepellent for *S.* Typhimurium.

Based on its function in *E. coli*, we hypothesized that both indole chemorepulsion and L-Ser chemoattraction for *S.* Typhimurium could be partly or fully mediated through Tsr (*Ortega et al., 2017*; *Yang et al., 2020*; *Burt et al., 2020*). We compared the chemotactic responses of the WT and *tsr* strains when exposed to sources of these effectors and found Tsr to be required for both chemorepulsion from indole and chemoattraction to L-Ser (*Figure 6E, F*). The canonical mode of chemoreceptor effector recognition involves binding of the effector to the periplasmic ligand-binding domain (LBD) (*Ortega et al., 2017*; *Gavira et al., 2020*), but the mechanism by which indole is sensed through Tsr in *Salmonella* has not been elucidated. We recently reported the first crystal structure of *S.* Typhimurium Tsr LBD, which clearly defines how the binding site recognizes the L-Ser ligand (PDB code: 8fyv), and we thought it unlikely indole can be accommodated at the same site (*Glenn et al., 2024*). To our knowledge, no prior study has tested whether the Tsr LBD binds indole directly, so we expressed and purified the LBD, corresponding to the soluble periplasmic portion, and performed isothermal titration calorimetry (ITC). These data show that no binding occurs between the Tsr LBD and indole (*Figure 6G*).

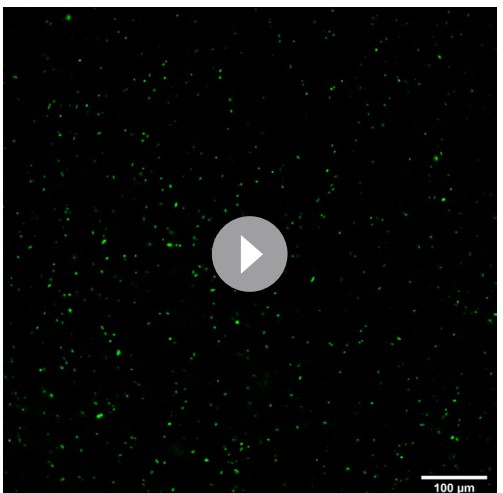

**Video 10.** Chemotactic response of *S.* Typhimurium IR715 to mixture of fecal effectors with 0.5× indole. Representative chemosensory injection rig assay (CIRA) experiment over 300 s. Viewable at: https://youtube.com/shorts/OqH0HE2rYIE?feature=share.

https://elifesciences.org/articles/106261/figures#video10

We next wondered if indole acts as an allosteric regulator of the LBD, possibly through interacting with the L-Ser-bound form or interfering with L-Ser recognition. To address these possibilities, we performed ITC of 50 µM Tsr LBD with L-Ser in the presence of 500 µM indole and observed a robust exothermic binding curve and $K_D$ of 5 µM, identical to the binding of L-Ser alone, which we reported previously (*Figure 6H*; *Glenn et al., 2024*). These data indicate that indole does not alter the Tsr LBD affinity for L-Ser. We conclude that Tsr senses indole through an atypical mechanism, which might either involve regulation through a solute-binding protein (*Yang et al., 2020*; *Matilla et al., 2021*), responsiveness to perturbation in the proton motor force (*Gupta et al., 2022*), or binding to a different region other than the periplasmic LBD. Our findings reveal that while indole acts as a chemorepellent for *S.* Typhimurium in isolation and is sensed through Tsr, its presence within fecal material mixed with other effectors is insufficient to elicit chemorepulsion.

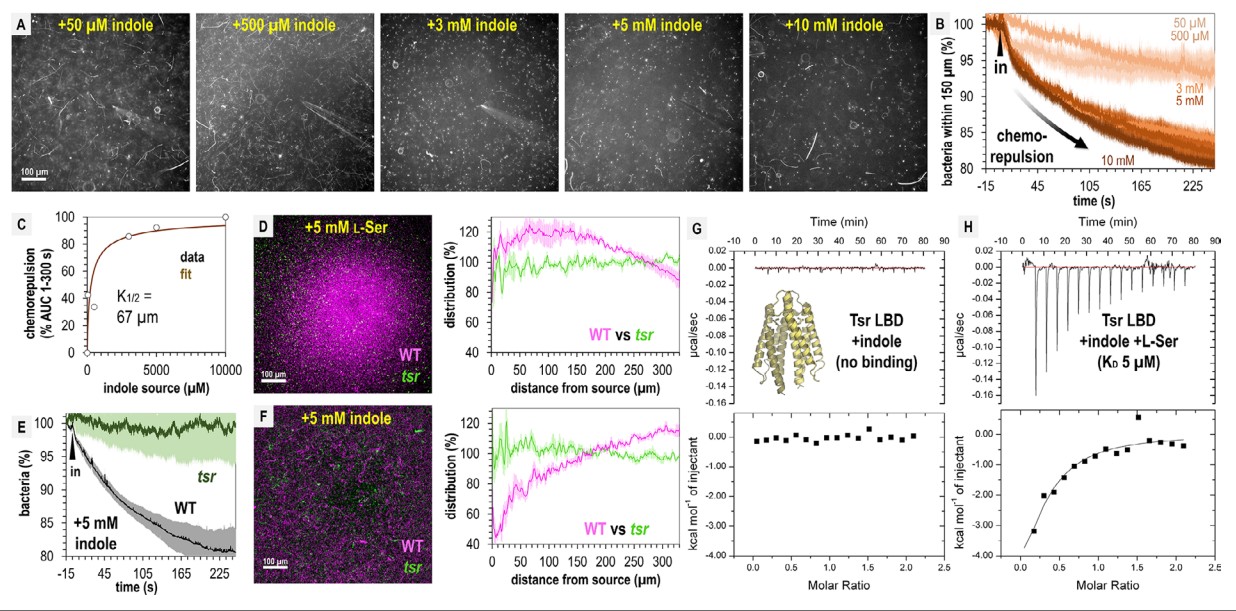

**Figure 6.** Tsr mediates indole chemorepulsion in *S.* Typhimurium. (**A**) Representative max projections of responses at 295–300 s of indole treatment. (**B**, **C**) Quantification of chemorepulsion as a function of indole concentration (*n* = 3–5). (**D–F**) Comparison of wildtype (WT) and *tsr* mutant responses to L-Ser or indole. (**E**) Quantification of the relative number of cells in the field of view over time following treatment with 5 mM indole for a competition experiment with WT and *tsr* (representative image shown in **F**). Data were collected at 30°C. (**G**, **H**) Isothermal titration calorimetry (ITC) experiments with 50 µM *S.* Typhimurium Tsr ligand-binding domain (LBD) and indole, or with L-Ser in the presence of 500 µM indole. Data are means and error bars are standard error of the mean (SEM, *n* = 3–5). AUC indicates area under the curve.

## Compromising between conflicting effector signals through chemohalation

Since Tsr mediates both chemoattraction to L-Ser and chemorepulsion from indole, we wondered at what threshold each response dominates, and how this behavior is regulated at the point of transition. To address these questions, we assessed responses to physiological mixtures of these effectors using 500 µM L-Ser and increasing concentrations of indole at L-Ser:indole molar ratios of 10:1, 1:1, or 1:10 (*Figure 7A–D*, *Video 11*). These experiments reveal a fascinating transition in the distribution of the pathogen population as a function of increasing chemorepellent, which occurs within minutes of exposure (*Figure 7A–D*, *Video 11*).

With only the chemoattractant present, the bacterial population organizes tightly around the effector source (*Figure 7A*, *Video 11*). When indole is introduced at a concentration 10-fold lower than L-Ser, the bacterial distribution still exhibits chemoattraction but becomes more diffuse (*Figure 7B*, *Video 11*). At a 1:1 ratio of chemoattractant and chemorepellent, a different population structure emerges, in which the swimming bacteria are attracted toward the source but form a halo around the treatment with an interior region of avoidance (*Figure 7C, E*, *Video 11*). When the concentration of indole is 10-fold higher than L-Ser, the bacteria exhibit a wider zone of avoidance (*Figure 7D, E*, *Video 11*). Interestingly, whereas 5 mM indole on its own induces strong chemorepulsion (*Figure 2— figure supplement 1I*, *Video 11*), the addition of 10-fold lower L-Ser effectively converts the behavior to a null response (*Figure 7D, E*, *Video 11*). This demonstrates that even at the highest concentrations of indole, *S.* Typhimurium might encounter in the gut, the presence of chemoattractant can override indole chemorepulsion.

The intermediate responses to opposing effector mixtures bear similarities to CIRA experiments with fecal material, some of which also exhibited a halo-like structure around the treatment source (*Figure 3*, *Videos 2 and 3*). Previous studies have also observed responses that are an intermediate behavior between chemattraction and chemorepulsion and have been referred to by a variety of names (*Livne and Vaknin, 2022*; *Englert et al., 2009*; *Huang et al., 2017*). There exists no consensus descriptor for taxis of this nature, and so we suggest expanding the lexicon with the term 'chemohalation', in reference to the halo formed by the cell population, and which is congruent with the

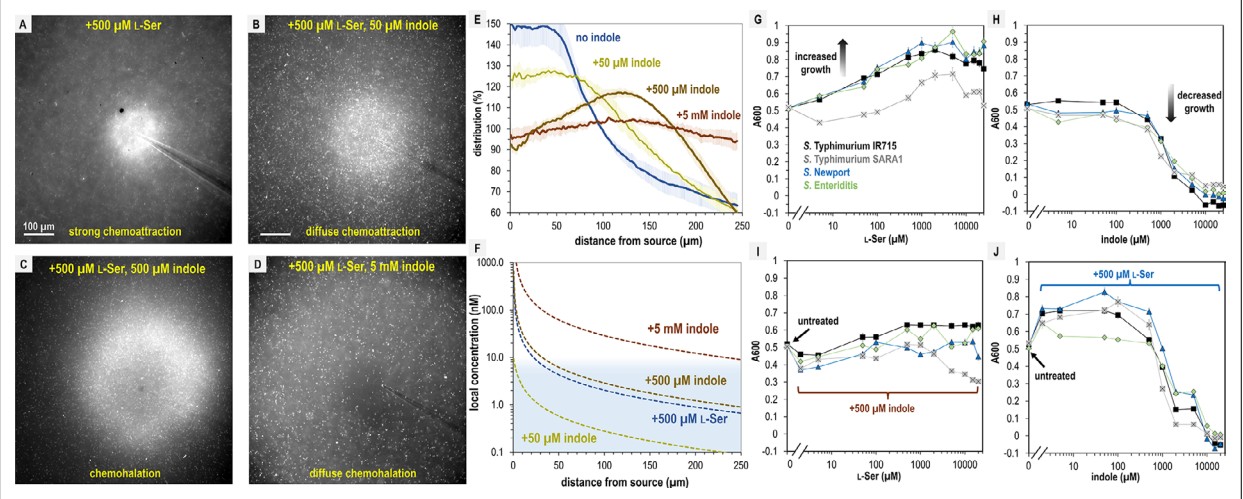

**Figure 7.** *S.* Typhimurium mediates distinct chemotactic responses based on the ratio of L-Ser to indole. (**A–D**) Representative max projections of responses to treatments of L-Ser and indole at 295–300 s, as indicated. (**E**) Relative bacterial distribution in response to treatments of 500 µM L-Ser and varying amounts of indole, from panels **A–D**, with the mean value normalized to 100%. Data were collected at 30°C. Data are means and error bars are standard error of the mean (SEM, *n* = 3–5). (**F**) Diffusion modeling of local effector concentrations based on sources of 5 mM indole (dark brown), 500 µM L-Ser (blue), 500 µM indole (light brown), and 50 µM indole (yellow) are shown as dashed lines. The approximate local concentration of indole that elicits a transition in chemotactic behavior is highlighted in light blue. (**G, H**) Bacterial growth as a function of L-Ser or indole, at the time point where the untreated culture reaches $A_{600}$ of 0.5. (**I, J**) Bacterial growth ± pretreatment with 500 µM indole or L-Ser, and increasing concentrations of indole or L-Ser, as indicated at the time point where the untreated culture reaches $A_{600}$ of 0.5. Data are means and error bars are standard error of the mean (SEM, *n* = 8–24). See also *Video 11*.

commonly used terms chemoattraction and chemorepulsion. We expect chemohalation is a compromise between the chemoattraction driven by L-Ser and the chemorepulsion driven by indole. Across these experiments, the interior zone of avoidance roughly corresponds to where the local concentration of indole is calculated to exceed 10 nM (*Figure 4E, F*).

## L-Ser enables resilience to indole-mediated growth inhibition

We questioned why non-typhoidal *Salmonella* are attracted to a biological solution with high amounts of indole, a chemical reported to inhibit bacterial growth (*Lee and Lee, 2010*; *Chimerel et al., 2012*; *Melander et al., 2014*). We examined how growth is affected by 0–25 mM indole or L-Ser in a background of minimal media (MM, Materials and methods). As expected, increasing amounts of the nutrient L-Ser provide a growth advantage for all *Salmonella* strains analyzed, with maximal benefit

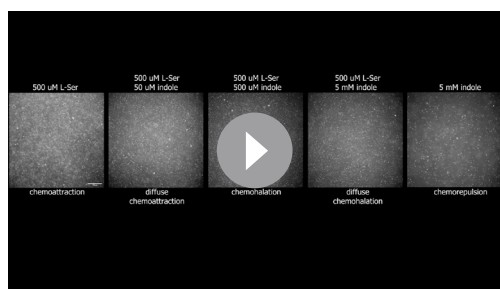

**Video 11.** Chemotactic response of *S.* Typhimurium IR715 to L-Ser and indole treatments. Representative chemosensory injection rig assay (CIRA) experiments with treatment sources as indicated, over 300 s. Viewable at: https://www.youtube.com/watch?v=bNQMqF2QMek.

https://elifesciences.org/articles/106261/figures#video11

achieved by approximately 500 µM (*Figure 7G*). Equivalent treatments with indole show tolerance up to approximately 1 mM, with growth inhibition occurring in the 1–5 mM range and lethality occurring at indole concentrations greater than 5 mM (*Figure 7H*). However, adding L-Ser in a background of 500 µM indole provides only a small growth enhancement (*Figure 7I*), and addition of 500 µM L-Ser increases tolerance for indole up to about 1 mM, above which indole toxicity is unavoidable (*Figure 7J*). It appears that the relative attraction to combinations of these effectors relates to their propensity to enhance or inhibit growth, with more permissive conditions eliciting a greater degree of chemoattraction. Overall, the bacteria still obtain growth benefits from L-Ser so long as the concentration of indole is under 1 mM.

## Discussion

Bacteria in the human gastrointestinal tract encounter complex chemical landscapes that contain both chemoattractants and chemorepellents (*Glenn et al., 2024*; *Matilla and Krell, 2018*; *Wishart et al., 2022*). However, chemotaxis responses are often studied in isolation, outside of their biological and ecological contexts, which can lead to an over- or underestimation of the roles specific interactions play in natural settings. In the present work, we contribute to an emerging understanding of how bacteria navigate conflicting chemotaxis stimuli and relate these chemotactic compromises to enteric infection and pathogen growth (*Livne and Vaknin, 2022*; *Huang et al., 2017*; *Moore et al., 2024*).

In this study, we show that despite the microbial metabolite indole being a strong chemorepellent in isolation (*Figure 6*), fecal indole is insufficient to elicit pathogen chemorepulsion, meaning that is pathogens do not swim away from fecal material, nor protect against cellular invasion in an explant model (*Figures 1–4*). Instead, it appears pathogens employ indole taxis as a means to regulate the magnitude of their attraction toward sources of intestinal nutrients (*Figure 5*). In vivo, we expect that the bacteria are attracted to indole-rich fecal material, and it is simply a matter of the degree of attraction and which sites are prioritized among those accessible to the invading pathogen. This finding revises our understanding of indole taxis during enteric infection, suggesting that, rather than impairing pathogen infection as others have proposed (*Yang et al., 2020*; *Kohli et al., 2018*), indole chemorepulsion serves a useful function for pathogens and enables them to integrate information about local microbial competitors into their chemotaxis responses. This, in turn, allows pathogens to prioritize niches with abundant nutrients and reduced microbial competition.

### Interpretations of explant infections and the functions of chemotaxis and Tsr

Our explant experiments can be thought of as testing whether a layer of effector solution is permissive to pathogen entry to the intestinal mucosa, and whether chemotaxis provides an advantage in transiting this chemical gradient to associate with, and invade, the tissue (*Figures 1C and 8*). This behavior is probabilistic, and given sufficient time, even chemotactic-deficient cells will contact the tissue. This is reflected in that all treatments showed substantial infection by all strains in terms of absolute colony-forming units (CFUs) isolated from homogenates (*Figure 1—figure supplement 1*). If we compare the probability of chemotaxis-mediated transit of the effector gradients we tested, the greatest is for fecal treatment, which among all treatments showed the highest degree of intracellular invasion (at 3 hr post-infection, *Figure 1*). Then, buffer, L-Ser, and L-Ser + indole treatments are similarly permissive, and chemotaxis enhances infection in these backgrounds as well (*Figures 1 and 8*). That chemotaxis provides an advantage in the buffer-treated background, without any added chemoeffector, is interesting and could simply be from effectors emitted from the host tissue (*Figure 8*). For instance, there is evidence that intestinal tissue, and host cells more broadly, can release L-Ser and other amino acids, particularly in the context of tissue injury which could be caused by *Salmonella* epithelial invasion in these experiments (*Glenn et al., 2024*; *Zhou et al., 2023*; *Kitamoto et al., 2020*; *Gül et al., 2023*). Altogether, chemotaxis enhances the transit of the effector gradients mentioned above to access the host tissue (*Figure 8*).

The explant system offers new insights into whether indole is protective against pathogen infection. First, indole treatment does negate the infection advantage conferred by chemotaxis, which was a unique effect among the treatments we tested (*Figures 1 and 8*). This is an interesting result, somewhat mirroring what others have seen in cell culture (*Kohli et al., 2018*), and indicates that the indole gradient does not increase the likelihood of transit for chemotactic cells. In assessing the total bacteria isolated from homogenates, we see no evidence that indole protects against infection, since the bacteria counts are prevalent and similar to other treatments, though this could be different at lower multiplicities of infection (*Figure 1—figure supplement 2*). Second, this effect is only observed with indole as the sole effector, but not when the same concentration of indole is present within fecal material, as it exists in the intestinal environment, or co-treatment with the fecal effector L-Ser (*Figures 1 and 8*). Thus, the loss of the chemotactic advantage observed with indole treatment is reversed in the presence of chemoattractant stimuli, suggesting that this effect is unlikely to occur in vivo. It is also worth noting that the residual effector concentrations experienced by the bacteria in the explant experiments were very low (*Figure 1C*), and so we do not think the effects we see are due to impacts on bacterial growth. It is unclear whether there would ever be a situation in vivo where indole

is the dominant effector, and so the behavior of bacteria swimming away from a source of pure indole may be somewhat artificial (*Figure 6*). These data, overall, do not support indole chemorepulsion as a mechanism of colonization resistance against pathogens, although indole is known to reduce virulence through other mechanisms (*Yang et al., 2020*; *Kumar and Sperandio, 2019*; *Kohli et al., 2018*; *Chimerel et al., 2012*).

## New insights into indole taxis from non-*E. coli* systems

Indole is a key regulator of enteric microbial communities, known to modulate motility and virulence, and is highly abundant in fecal matter due to the metabolic activity of the microbiota (*Yang et al., 2020*; *Li et al., 2021*; *Lee and Lee, 2010*; *Gupta et al., 2022*; *Ye et al., 2022*; *Kumar and Sperandio, 2019*; *Kohli et al., 2018*; *Nikaido et al., 2012*). While *E. coli* has served as an important model system for elucidating the mechanisms of indole chemorepulsion (*Supplementary file 1*; *Yang et al., 2020*; *Gupta et al., 2022*; *Kohli et al., 2018*; *Bansal et al., 2007*), no prior work has examined how indole sensing is integrated alongside multiple other intestinal effectors, nor whether these behaviors are conserved across clinical isolates of disease-causing species. Here, we address these gaps by providing confirmatory evidence for some earlier predictions and evidence that challenges others, refining our understanding of how indole influences pathogen behavior through chemotaxis within the intestinal environment.

Perhaps our most striking finding is that fecal material, the native reservoir of the strong chemorepellent indole, does not elicit chemorepulsion in the form of bacteria swimming away (*Figures 2–4*). Instead, a representative panel of diverse *Enterobacteriaceae* pathogens and pathobionts exhibited fecal attraction (*Figures 3 and 4*), demonstrating that enteric species associated with disease are undeterred by indole in its natural context, that is, when mixed with other fecal chemoattractants. Through analyses showing that indole chemorepulsion is easily overridden by the presence of intestinal nutrients, we surmise that the benefits associated with fecal material typically outweigh the deleterious effects of indole (*Figures 5 and 7*). This conclusion is further supported by growth analyses indicating that *Salmonella* tolerates indole when sufficient nutrients are available (*Figure 7*).

We used *Salmonella* Typhimurium as a model to dissect the mechanisms underlying fecal attraction and indole sensing. Regarding the latter, our findings largely confirm previous studies in *E. coli*, showing that the chemoreceptor Tsr is required for indole taxis (*Figure 6*, *Supplementary file 1*; *Yang et al., 2020*; *Gupta et al., 2022*; *Bansal et al., 2007*). However, we do add some new dimensions to understanding indole taxis. First, for *E. coli* the involvement of Tar in indole sensing has been reported (*Yang et al., 2020*), but we see no equivalent function for *S.* Typhimurium. Yet, we previously showed *S.* Typhimurium WT and *tsr* are both readily attracted to L-Asp, supporting the presence of a functional Tar under the same experimental conditions as we test here (*Figure 6*; *Glenn et al., 2024*). We do not know the reason for this outcome, but note different assays were used, and this could also reflect variation between the chemotaxis systems of these two bacteria. Second, while Tsr serves as the sensor for indole in *Salmonella*, it is also a key mediator of fecal attraction through its role in sensing L-Ser, which is also abundant in fecal material (*Figures 1–3 and 6*). Third, we are the first to visualize and quantify the rapid temporal dynamics of indole chemorepulsion (*Figures 5 and 6*, *Figure 2—figure supplement 1I*, *Video 11*). For responses to pure indole, a clear zone of avoidance around the treatment appears within 10 s of exposure, much faster than chemoattraction to L-Ser, suggesting the cells have the ability to rapidly flee deleterious conditions (*Figure 6*, *Figure 2—figure supplement 1I*, *Video 11*). Lastly, we also investigated whether indole sensing occurs through the canonical chemoreceptor mechanism of direct binding to the Tsr LBD. Our data show it does not, nor does indole antagonize or inhibit L-Ser binding to the LBD (*Figure 6*). While these findings do not resolve the molecular mechanism of indole sensing, they eliminate two plausible models that, to our knowledge, have not been previously tested. Overall, our data support the hypothesis that Tsr employs a non-canonical mechanism to sense indole (*Yang et al., 2020*; *Gupta et al., 2022*; *Matilla et al., 2021*).

Having confirmed the role of Tsr in mediating indole chemorepulsion in *S.* Typhimurium, and shown it to function similarly as in *E. coli* (*Yang et al., 2020*; *Livne and Vaknin, 2022*; *Englert et al., 2009*), and having demonstrated that diverse *Enterobacteriaceae* are attracted to indole-rich fecal material, we expect that the behaviors described here are representative of the many enteric species that possess Tsr orthologs, which we mapped in a previous study (*Glenn et al., 2024*). As we report here,

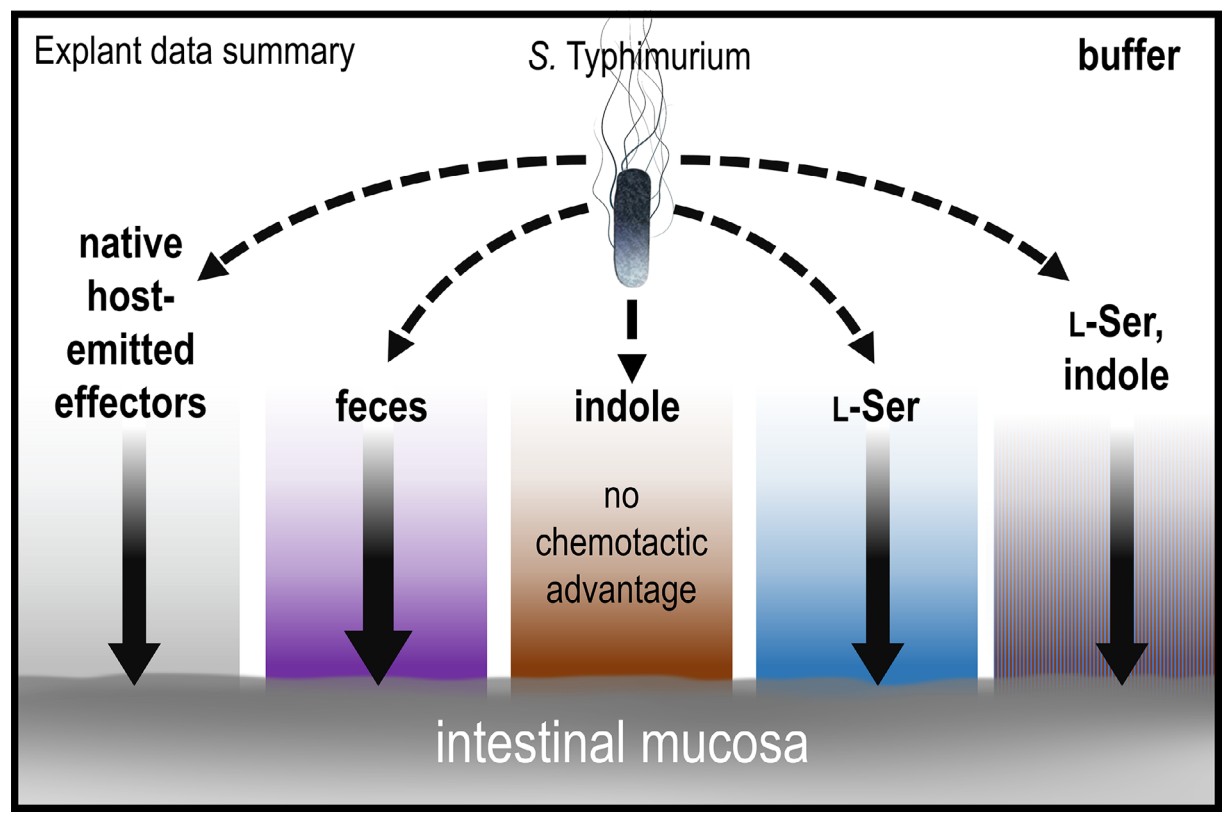

**Figure 8.** Model and summary for explant infection data. Based on analyses in this study, we provide this summary of the role of chemotaxis in mediating infection advantages for different tissue pretreatments. The strength of chemotactic advantage for transiting each chemical gradient and accessing the host tissue is indicated by the width of the solid black arrows. The baseline level of chemotactic advantage seen in buffer treatments may be from effectors emitted from the host tissue (gray gradient). Other gradients containing fecal chemoattractants show a similar level of chemotactic advantage, with the highest being for fecal treatment (purple). Only soaking the tissue in pure indole results in a chemical gradient for which chemotaxis and Tsr do not provide an infection advantage. Note that bacteria are exposed only to low concentrations of residual effectors that remain after the tissue is soaked and then transferred to 300 µl of buffer for infection; they are not immersed in the more concentrated effector solution. See also *Figure 1*, *Figure 1—figure supplement 1*.

there does seem to be a large variety in the magnitude of fecal attraction by different 'wild' enteric pathogens, which could reflect adaptations to different host intestinal environments and microbiota communities and may influence pathogenesis (*Figures 3 and 4*). While foundational insights into indole taxis have come from model bacterial systems, continued progress in understanding the role of chemotaxis in human disease will benefit from extending such analyses to a broader range of clinically relevant bacterial species and strains.

### Function of indole taxis in enteric invasion

In the context of non-typhoidal *Salmonella* infections, it is clear that complex relationships exist between chemotactic sensing of effectors, bacterial growth, and invasion (*Figure 8*). In addition to the factors we have investigated, it is already well established in the literature that the vast metabolome in the gut contains many chemicals that modulate *Salmonella* cellular invasion, virulence, growth, and pathogenicity (*Antunes et al., 2014*; *Peixoto et al., 2017*; *Lamas et al., 2019*). As it pertains specifically to sensing the opposing effectors L-Ser and indole, we propose that Tsr directs bacteria toward the highest ratio of attractant to repellent accessible in the local environment, with fine-tuning of navigation occurring through regulation of the magnitude of attraction and chemohalation. In addition to sensing these two effectors, our data indicate that fecal attraction involves other stimuli, including L-Asp sensing through Tar and sugar sensing through Trg (*Figure 5*). Ultimately, the dual sensing of opposing effectors by Tsr serves to improve pathogen fitness through colonizing niches rich in

nutrients, signaled by local L-Ser concentrations, and seeking niches with low microbial competition, indicated by local indole concentrations.

## Navigating contradictory stimuli in nature

The scenario we investigated here of *S.* Typhimurium encountering high concentrations of opposing chemotactic stimuli in the intestinal environment is just one example of the complex chemical landscapes that bacteria navigate in nature. To better understand the 'decision-making' process underlying chemotaxis in the presence of conflicting effectors, we examined physiological mixtures of the fecal metabolites indole and L-Ser and recorded a series of real-time videos capturing behavioral transitions as a function of effector concentration (*Figure 7*, *Video 11*). These videos reveal that, upon sensing conflicting stimuli, the bacterial population structure rapidly evolves based on the attractant-to-repellent ratio, displaying a spectrum of behaviors: chemoattraction, diffuse chemoattraction, chemohalation, diffuse chemohalation, and chemorepulsion (*Figure 7*, *Video 11*).

These dynamic, micron-scale chemohalation patterns reflect a behavioral compromise between attraction and repulsion and would be difficult or impossible to detect without live imaging, which may explain why they were previously unappreciated in binary models of chemotaxis (*Adler and Tso, 1974*). In fact, many chemotaxis assays that use indirect methods of quantification, such as growth, would not be able to distinguish between chemoattraction and chemohalation since they both involve an increase in bacteria over time. To be clear, we suggest chemohalation as a new term to generally describe intermediate chemotaxis responses to conflicting stimuli that are neither chemoattraction nor chemorepulsion, but others have contributed to studying how chemotaxis functions in confounding chemical landscapes (*Livne and Vaknin, 2022*; *Englert et al., 2009*; *Huang et al., 2017*). The chemohalation responses reported here most closely resemble the 'trade-off' response previously described in *E. coli* exposed to attractant–repellent mixtures (*Livne and Vaknin, 2022*). Interestingly, that study also described a 'bet-hedging' response, in which a subpopulation remained attracted despite the presence of a chemorepellent, but we did not observe this behavior in our system (*Livne and Vaknin, 2022*).

In addition to our work here, there are other examples of chemohalation responses to complex biological stimuli of the gastrointestinal environment. Recently, we reported on *Enterobacteriaceae* chemotactic sensing of blood serum, which bacteria encounter during intestinal bleeding events, and those responses appear to involve chemohalation (*Glenn et al., 2024*). Chemohalation is also seen in the case of the gastric pathogen *Helicobacter pylori* responding to mixtures of urea, a chemoattractant, and acid, a chemorepellent, conflicting effectors it encounters near the stomach mucosa (*Huang et al., 2017*; *Huang et al., 2015*). The functional significance of chemohalation remains to be understood, but could be a method of fine-tuning colonization bias such that nutrients can be acquired while not approaching too closely to a deleterious stimulus. Continuing to investigate chemohalation behaviors and understanding how they coordinate bacterial colonization may provide important insights into how chemotaxis confers fitness advantages in natural environments.

## Limitations of this study

This study provides insights into the roles of chemotaxis in directing the behaviors of *Enterobacteriaceae* species in response to fecal material and indole; however, several limitations should be considered. First, our analyses of pathogen enteric infection were performed using swine colonic explants, which do not fully recapitulate the complexity of in vivo infection dynamics in the human gut. While explant assays offered insights into the relationship between chemotaxis and tissue colonization, these experiments exhibited variability. To mitigate this, we used multiple tissue sections from a single animal to improve experimental consistency. However, this approach limits our ability to assess how inter-host variability might influence bacterial responses. Future studies using distal ileum tissue, a major site of *S.* Typhimurium cellular invasion known to contain distinct chemical features, may provide further insight into the functions of indole taxis during infection (*Chowdhury et al., 2023*). Another experimental limitation is the difference in timescales between our assays. Chemotaxis experiments were conducted over approximately 5 min, whereas tissue explant experiments required several hours for significant differences in colonization and cellular invasion to be observed. Thus, there are effects from chemotactic adaptation and replication that we do not elucidate here. Lastly, while we confirmed that non-typhoidal *Salmonella* are attracted to human fecal material, we

only determined the dependency on Tsr and sensing of L-Ser in our model strain (IR715). Although we predict that other *Enterobacteriaceae* also use Tsr for fecal attraction, this remains uncertain without targeted genetic analyses in each strain background.

## Materials and methods

### Key resources table

| Reagent type (species) or resource | Designation | Source or reference | Identifiers | Additional information |
|---|---|---|---|---|
| Strain, strain background (*Salmonella enterica* serovar Typhimurium) | *Salmonella enterica* serovar Typhimurium | *Rivera-Chávez et al., 2013* | IR715 | Nalidixic acid-resistant derivative of ATCC 14028 |
| Strain, strain background (*S. enterica* Typhimurium) | *S. enterica* Typhimurium *cheY* mutant | *Rivera-Chávez et al., 2013* | FR13 | IR715 *cheY*::Tn10 (Tet[R]) |
| Strain, strain background (*S. enterica* Typhimurium) | *S. enterica* Typhimurium *tsr* mutant | *Rivera-Chávez et al., 2013* | FR4 | IR715 *tsr*::pFR3 (Cm[R]) |
| Strain, strain background (*S. enterica* Typhimurium) | *S. enterica* Typhimurium *invA* mutant | *Thiennimitr et al., 2011* | SW399 | IR715 *invA*::pSW127 (Carb[R]) |
| Strain, strain background (*S. enterica* Typhimurium) | *S. enterica* Typhimurium Clinical Isolate | *Beltran et al., 1991* | SARA1 | Isolated from patient in Mexico |
| Strain, strain background (*S. enterica* Newport) | *S. enterica* Newport Clinical Isolate | *Shariat et al., 2013* | M11018046001A | Isolated from patient in PA, USA |
| Strain, strain background (*S. enterica* Enteriditis) | *S. enterica* Enteriditis Clinical Isolate | *Shariat et al., 2013* | 05E01375 | Isolated from patient in PA, USA |
| Strain, strain background (*Citrobacter koseri*) | *C. koseri* Clinical Isolate | ATCC | BAA-895 | Human Clinical Isolate |
| Strain, strain background (*Enterobacter cloacae* subsp. *cloacae*) | *Enterobacter cloacae* subsp. *Cloacae* clinical isolate | ATCC | 13047 | Human Clinical Isolate |
| Strain, strain background (*Escherichia coli*) | *E. coli* Clinical Isolate | ATCC | 11775 | Human Clinical Isolate |
| Strain, strain background (*Escherichia coli*) | BL21(DE3) | Millipore Sigma | 70954–3 | Electrocompetent cells |
| Biological sample (*Homo sapiens*) | Human feces | Lee Biosolutions | 991–18 | See method details |
| Recombinant DNA reagent | XS Plasmid expressing sfGFP | *Wiles et al., 2018* | pXS-sfGFP | pGEN-mcs with a modular sfGFP expression scaffold (Amp[R]) |
| Recombinant DNA reagent | XS Plasmid expressing mPlum | *Wiles et al., 2018* | pXS-mPlum | pGEN-mcs with a modular mPlum expression scaffold (Amp[R]) |
| Peptide, recombinant protein | *Se*Tsr LBD | *Glenn et al., 2024* | | See 'Method details' |

All methods were carried out in accordance with relevant guidelines, regulations, and state and federal law. Experimental protocols were approved by the Institutional Biosafety Committee (IBC) of Washington State University (#1372).

### Bacterial strains and growth conditions

Bacterial strains and plasmids used in this study are listed in the Key Resources Table. As previously described (*Glenn et al., 2024*), bacteria intended for chemotaxis assays were grown overnight in tryptone broth (TB) with antibiotic selection, as appropriate. Motile bacteria were prepared with a 1:1000 back-dilution and grown shaking for approximately 4 hr at 37°C to reach $A_{600}$ of 0.5. Cells were centrifuged, washed, and resuspended in a chemotaxis buffer (CB) containing 10 mM potassium phosphate (pH 7), 10 mM sodium lactate, and 100 µM EDTA to $A_{600}$ of 0.2 and rocked gently at the temperatures indicated in figure legends until fully motile, typically 1–2 hr. For in vitro growth analyses, cultures were grown overnight in Lysogeny Broth (LB) at 37°C. Subsequently, 5 µl of $A_{600}$ 2.0 cells were used to inoculate 200 µl of MM, containing 47 mM $Na_2HPO_4$, 22 mM $KH_2PO_4$, 8 mM NaCl, 2 mM $MgSO_4$, 0.4% glucose (wt/vol) 11.35 mM $(NH_4)_2SO_4$, 100 µM $CaCl_2$, and L-Ser and/or indole at the described concentrations, and cultured in a 96-well microtiter plate. Cultures were grown at 37°C and monitored by $A_{600}$ readings at 5-min intervals.

## Chemosensory injection rig assay

CIRA was performed as described previously (*Glenn et al., 2024*). Briefly, an Eppendorf Femtotip 2 microcapillary containing the treatment of interest was lowered into a pond of 50 µl of motile cells using a Sutter micromanipulator. An injection flow of effector into the pond at approximately 300 fl per minute was achieved using a Femtojet 4i set to $P_c$ 35. Solubilized fecal treatments were prepared by dissolving 1 g of commercially obtained human feces (Lee Biosolutions) in 10 ml of CB. The solution was clarified by centrifugation at 10,000 × g for 20 min, followed by sterile filtration through a 0.2 µm filter. Treatment solutions of indole and L-Ser were also diluted into CB and sterile-filtered before application. Videos were recorded using an inverted Nikon Ti2 microscope with heated sample chamber at 37°C.

## CIRA microgradient modeling

Modeling the microgradient generated through CIRA was performed as described earlier (*Glenn et al., 2024*), based on the continual injection and diffusion of an effector from a fixed-point source. Briefly, diffusion is modeled as a 3D process where the diffusing substance is gradually and continuously introduced at a fixed point within a large surrounding fluid volume. The substance is prepared at a concentration of $M_s$ (typically between 0.5 µM and 5 mM) and injected at a volume rate of $Q = 305.5$ fl/min. The species then diffuses into the ambient fluid with a diffusion constant $D$:

$$C\left(r,t\right) = \frac{q}{4\pi Dr}erfc\frac{r}{2\sqrt{Dt}}.$$

Here, $r$ is the distance from the point source, $t$ is the time from initial injections, $q$ is the injection rate of the species (equal to $M_sQ$), and $C$ is the species concentration. In earlier work (*Glenn et al., 2024*), we reported using fluorescent dye that the concentrations predicted by this model appear to be accurate within 5% in the range of 70–270 µm from the source, whereas at distances less than 70 µm the measured concentrations are about 10% lower than predicted. At the point where the effector treatment is injected into the larger volume, the local concentration drops precipitously, hence why the concentration reported at distance 0 is not that of the concentration within the microcapillary.

## ITC ligand-binding studies

Purification of *S.* Typhimurium Tsr LBD was performed as described previously (*Glenn et al., 2024*). ITC experiments were performed using a Microcal ITC200 instrument (GE Healthcare). Either 500 µM indole or L-Ser was titrated in 2.5 µl injections into a 200-µl sample cell containing 50 µM Tsr LBD. For the indole/L-Ser competition experiment, 500 µM indole was added to both the titrant and sample cell, thus providing a constant excess background concentration of indole. For all experimental conditions, blank titrations were also collected in which indole or L-Ser was titrated into a cell containing buffer alone. All experiments were performed using thoroughly degassed samples at 25°C in 50 mM Tris, 150 mM NaCl, 1 mM EDTA, pH 7.5. The reference power was set to 5 µcal/s. The resulting power curves were integrated using the Origin analysis software included with the instrument. The heat of dilution was subtracted from each point using the blank. A single-site binding model was then fit to the data, floating parameters describing the binding enthalpy ($\Delta H$), equilibrium constant ($K_D$), and apparent binding stoichiometry ($n$). The instrument software was used for this purpose.

## Quantification of indole and serine in human fecal samples

Solubilized human feces was prepared as described above for CIRA and analyzed by mass spectrometry to determine the molar serine content as a service through the University of Washington Mass Spectrometry Center. This measurement reflects total serine, of which close to 100% is expected to be L-Ser (*Glenn et al., 2024*). As described in earlier work, the indole content of solubilized human fecal samples was determined using a hydroxylamine-based calorimetric assay with purified indole as a reference and standard (*Sugihara et al., 2022*).

## Explant infection assays

Swine intestinal tissue was acquired from the descending colon of an 8-week-old animal, pursuant to animal protocol ASAF #7128, approved through the Washington State University IACUC. Before infection, an approximately 20 by 20 mm piece of swine intestinal explant tissue was gently washed

with PBS to remove fecal matter. Next, the tissue section was bathed in chemoeffector solution (solubilized human fecal matter (Lee Biosolutions)), a mixture of 338 µM L-Ser and 862 µM indole, 338 µM L-Ser alone, 862 µM indole alone, or CB in a 6-well tissue culture plate (Celltreat) and incubated at 4°C for 1 hr. Then, tissue was transferred to a 35-mm Mattek dish where the luminal side of the tissue was exposed to a bacterial solution containing a 1:1 mixture (~$10^9$ CFU each) of WT *S.* Typhimurium IR715 and either the isogenic *tsr* or *cheY* mutant, suspended in CB at a volume of 300 µl. The tissue was then incubated in the dish with the competing bacteria at 37°C and 5% $CO_2$ for 1, 3, or 6 hr. After, half of the tissue was transferred into screw cap tubes containing 500 µl LB media and 5–10 2.3 mm zirconia beads (BioSpec Products) on ice and homogenized using a Bead Mill 24 (Fisher Scientific) at 6.5 m/s for 60 s, repeated four times. To enumerate the 'invaded' bacteria, the other half of the tissue was washed in PBS and incubated in PBS containing 100 µg/ml gentamicin for 1 hr at 37°C and 5% $CO_2$, then washed twice in PBS, as done previously (*Sharma and Puhar, 2019*; *Chandra, 2022*; *Rivera Calo et al., 2024*). The homogenization process was then repeated for the gentamicin-treated tissue. CFUs were enumerated by plating 10-fold dilutions on LB agar plates containing the appropriate antibiotic (*Sharma and Puhar, 2019*; *Ben-David and Davidson, 2014*). Competitive index (CI) values were calculated by dividing the number of mutant CFUs by the number of WT CFUs for each treatment and time point (*Macho et al., 2007*; *Auerbuch et al., 2001*).

## Quantification of CIRA data

Videos of chemotactic responses were quantified as described previously (*Glenn et al., 2024*). The number of cells in each frame was calculated by determining a fluorescence intensity ratio per cell for frames pretreatment and extrapolated using the 'plot profile' function of ImageJ. The distribution of the bacteria was calculated using the Radial Profile ImageJ plugin. Local background subtraction was performed based on experiments with the non-chemotactic *cheY* strain to control for autofluorescence in solubilized fecal samples.

## Statistical analyses

CIs for explant experiments were calculated for each treatment group at each time point. Log-transformed CI values were obtained by taking the logarithm ($log_{10}$) of the original CI measurements. These log-transformed values were then subjected to statistical analysis. First, a one-sample *t*-test was performed to determine whether the mean of the log-transformed CIs significantly differed from zero. In cases where the assumption of normality was violated, the non-parametric Wilcoxon rank sum test was applied as an alternative. Effect size was assessed using Cohen's *d* and calculated using the same log-transformed CIs. To determine p-values between total and invaded populations at 3 and 6 hr and for comparing relative bacteria % within 500 µm of the treatment source in *Figures 5 and 6*, unpaired *t*-tests were employed.

## Acknowledgements

Funding for this research was provided by NIAID through awards 1K99AI148587 and 4R00AI148587-03, and funding from the College of Veterinary Medicine at Washington State University to AB. Bacterial strains were provided by Nikki Shariat (University of Georgia, Athens), Nkuchia Mikanatha and Pennsylvania NARMS and GenomeTrakr Programs, and Andreas Bäumler (University of California, Davis). All research on human and animal samples was performed in accordance with, and approval of, the Institutional Biosafety Committee and Institutional Animal Care and Use Committee at Washington State University.

---

## Additional information

### Competing interests
Arden Baylink: AB owns Amethyst Antimicrobials, LLC (Pullman, WA). The other authors declare that no competing interests exist.

## Funding

| Funder | Grant reference number | Author |
|--------|------------------------|--------|
| National Institute of Allergy and Infectious Diseases | 1K99AI148587 | Arden Baylink |
| National Institute of Allergy and Infectious Diseases | 4R00AI148587-03 | Arden Baylink |

The funders had no role in study design, data collection, and interpretation, or the decision to submit the work for publication.

## Author contributions

Kailie Franco, Conceptualization, Formal analysis, Investigation, Methodology, Writing – original draft, Writing – review and editing; Zealon Gentry-Lear, Michael Shavlik, Investigation, Writing – review and editing; Michael J Harms, Resources, Supervision, Investigation, Methodology, Writing – review and editing; Arden Baylink, Conceptualization, Data curation, Formal analysis, Supervision, Funding acquisition, Investigation, Visualization, Methodology, Writing – original draft, Writing – review and editing

## Author ORCIDs

Kailie Franco ⓘ http://orcid.org/0000-0003-4719-225X
Michael J Harms ⓘ https://orcid.org/0000-0002-0241-4122
Arden Baylink ⓘ https://orcid.org/0000-0001-5522-769X

Reviewer #1 (Public review): https://doi.org/10.7554/eLife.106261.3.sa1
Reviewer #3 (Public review): https://doi.org/10.7554/eLife.106261.3.sa2
Author response https://doi.org/10.7554/eLife.106261.3.sa3

# Additional files

## Supplementary files

Source data 1. CFUs from explant infection assays.

MDAR checklist

Supplementary file 1. Summary of prior studies related to indole chemotaxis.

## Data availability

Source data for the chemotaxis videos can be downloaded at https://public.vetmed.wsu.edu/Baylink/Franco-et-al_eLife-2025. Experimental conditions are noted in the 'README.txt' files. Because the raw video files reported in this study sometimes exceed 20 gigabytes each, totaling approximately 3 terabytes for all the videos in this study, it is not feasible to store all the video source files in a public repository. However, these data are available upon request from the corresponding author, Arden Baylink, at arden.baylink@wsu.edu. Any of the data presented in our work will be shared with requesters promptly and without restriction. The video files are supplied in .avi or .nd2 format and can easily be viewed using the free ImageJ software, available here: https://imagej.net/ij/download.html. No other specialized software or code is needed to view or analyze the data. Source data in the form of disaggregated colony formation unit enumeration for the explant infection studies are provided in *Source data 1*.

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
