## [Editor Report · eLife Assessment]

In this manuscript, Franco and colleagues present **compelling** evidence that fecal extracts containing high concentrations of indole, a known repellent, enhance rather than protect against invasion of colonic tissue by Salmonella. The authors describe **important** findings that lead to the conclusion that the competing effects of attractants present in fecal matter, including L-serine, also sensed by the Tsr chemoreceptor that senses indole, override the repulsive effect of indole.

---

## [Referee Report · Reviewer #1 (Public review)]

Summary:

The study shows, perhaps surprisingly, that human fecal homogenates enhance the invasiveness of Salmonella typhimurium into cells of a swine colonic explant. This effect is only seen with chemotactic cells that express the chemoreceptor Tsr. However, two molecules sensed by Tsr that are present at significant concentrations in the fecal homogenates, the repellent indole and the attractant serine, do not, either by themselves or together at the concentrations in which they are present in the fecal homogenates, show this same effect. The authors then go on to study the conflicting repellent response to indole and attractant response to serine in a number of different in vitro assays.

Strengths:

The demonstration that homogenates of human feces enhance the invasiveness of chemotactic Salmonella Typhimurium in a colonic explant is unexpected and interesting. The authors then go on to document the conflicting responses to the repellent indole and the attractant serine, both sensed by the Tsr chemoreceptor, as a function of their relative concentration and the spatial distribution of gradients.

Weaknesses:

The authors do not identify what is the critical compound or combination of compounds in the fecal homogenate that gives the reported response of increased invasiveness. They show it is not indole alone, serine alone, or both in combination that have this effect, although both are sensed by Tsr and both are present in the fecal homogenates. Some of the responses to conflicting stimuli by indole and serine in the in vitro experiments yield interesting results, but they do little to explain the initial interesting observation that fecal homogenates enhance invasiveness.

---

## [Referee Report · Reviewer #3 (Public review)]

Summary:

In this manuscript, Franco and colleagues describe careful analyses of Salmonella chemotactic behavior in the presence of conflicting environmental stimuli. By doing so, the authors describe that this human pathogen integrates signals from a chemoattractant and a chemorepellent into an intermediate "chemohalation" phenotype.

Strengths:

The study was clearly well-designed and well-executed. The methods used are appropriate and powerful. The manuscript is very well written, and the analyses are sound. This is an interesting area of research, and this work is a positive contribution to the field.

Weaknesses:

No significant weaknesses noted.

---

## [Author Response]

The following is the authors’ response to the original reviews

We thank the reviewers and editors for their careful consideration of our work and pointing out areas where the current version lacked clarity or necessary experiments. Based on the reviews we have made the following significant changes to the revised version:

(1) Revised the text to focus on the distinct pathogen responses to indole in isolation versus fecal material.

We believe the key takeaway from this work is that the native context of a given effector, in this case indole, can elicit markedly different bacterial responses compared to the pure compound in isolation. This is because natural environments contain multiple, often conflicting, stimuli that complicate predictions of overall chemotactic behavior. For example, while indole has been proposed to mediate chemorepulsion and contribute to colonization resistance against enteric pathogens, our findings challenge this model. We provide evidence that feces, the intestinal source of indole, actually induces attraction, and that indole taxis may in fact benefit the pathogen through prioritizing niches with low microbial competition. Put another way, the biological reservoir of indole, fecal material, generates an attraction response but indole regulated the degree of attraction.

Most current understanding of chemotaxis is based on responses to individual, purified effectors. Our study highlights the need to investigate chemotactic responses in the presence of native mixtures, which better reflect the complexity of natural environments and may reveal new functional insights relevant for disease.

Reviewer comments indicated that these core points above were not clearly conveyed in the previous version, and that the manuscript's logical flow needed improvement. In this revised version, we have substantially rewritten the text and removed extraneous content to sharpen the focus on these central findings. We have also aligned our discussion more closely with the experimental data. While we appreciated the reviewers’ thoughtful suggestions, we chose not to expand on topics that fall outside the scope of our current experiments.

(2) Provide new chemotaxis data with mixtures of fecal effectors (Fig. 5).

Related to the above, the reviewers and editors brought up concerns that our discovery of pathogen fecal attraction was underexplored. Although we showed Tsr to be important for mediating fecal attraction, even the tsr mutant showed attraction to a lesser degree, and the reviewers noted that we did not identify what other fecal attractants could be involved.

Fecal material is a complex biological material (as noted by Reviewer 3) and contains effectors already characterized as chemoattractants and chemorepellents. It would be ideal to be able to perform some experiment where individual effectors are removed from fecal material and then quantify chemotaxis. We considered methods to do this but ultimately found this approach unfeasible. Instead, we employed a reductionist approach and developed a synthetic approximate of fecal material containing a mixture of known chemoeffectors at fecal-relevant concentrations (Fig. 5). We used this defined system as a way to test the specific roles of the Tsr effectors L-Ser (attractant) and indole (repellent) in relation to glucose, galactose, and ribose (sensed through the chemoreceptor Trg), and L-Asp (sensed through the chemoreceptor Tar). We chose these effectors as they have reasonable structure-function relationships established in prior work, and had information available about their concentrations in fecal material. We present these data as a new Figure 5, and also provide videos clearly showing the responses to each treatment (Movies 7-10).

This defined system provided several new insights that help understand and model indole taxis amidst other fecal effectors. First, the complete effector mixture, like fecal treatment, elicits attraction. Second, L-Ser is able to negate indole chemorepulsion in cotreatments of the two effectors, and also other chemoattractants in the absence of L-Ser also negate this repulsion, albeit to a lesser degree, helping to explain why the tsr mutant still shows attraction to fecal material. Lastly, we also show that the degree of attraction in this system is controlled by indole, with mixtures containing greater indole showing less attraction. We feel this is an important addition to the study because it provides a new view on how indole-taxis functions in pathogen colonization; rather than causing the pathogen to swim away (like pure indole does) indole helps the pathogen rank and prioritize its attraction to fecal effector mixtures, biasing navigation toward lower indolecontaining niches.

We also acknowledge that this defined system does not capture all possible interactions. Indeed, there are even a few chemoreceptors in Salmonella for which the sensing functions remain poorly understood. Nonetheless, we believe the data offer mechanistic context for understanding fecal attraction and suggest that factors beyond Tsr, L-Ser, and indole also contribute to the observed behaviors, aligning with other data we present.

(3) Provide new data that show that *E. coli* MG1655, and disease-causing clinical isolate strains of the Enterobacteriaceae Tsr-possessing species *E. coli*, Citrobacter koseri, and Enterobacter cloacae exhibit fecal attraction (Fig. 4).

An important new finding from this study is our direct test of whether indole-rich fecal material elicits repulsion. Contrary to expectations, given that for *E. coli* indole is a wellcharacterized strong chemorepellent, we show that fecal material instead elicits attraction in non-typhoidal Salmonella.

Reviewers raised the question of whether our observations regarding indole taxis and attraction to indole-rich feces in Salmonella are similar or relevant to *E. coli*. While a full dissection of indole taxis in *E. coli* is beyond the scope of this study and has been the focus of extensive prior research, we sought to address this point by examining whether other enteric pathogens respond similarly to the native indole reservoir, fecal material. To this end, we present new data demonstrating that, like *S. Typhimurium*, *E. coli* and other representative enteric pathogens and pathobionts possessing Tsr are also attracted to indole-rich feces (Fig. 4, Movies 4–6, Fig. S4).

Notably, these new results represent some of the first characterizations of chemotactic behavior in the clinical isolates we examined, including *E. coli* NTC 9001 (a urinary tract infection isolate), Citrobacter koseri, and Enterobacter cloacae, adding another element of novelty to this work.

(4) Repeated all of the explant Salmonella Typhimurium infection studies and added a new experimental control competition between WT and an invasion-deficient mutant (invA).

Although our new colonic explant system was noted as a novelty and strength of this work, it was also seen as a weakness in that some of the results were surprising and difficult to link to chemotactic behavior. Reviewer 3 also brought up the need to be clear about our usage of the term ‘invasion’ in reference to S. Typhimurium entering nonphagocytic host cells, and requested we test an invasion-inhibited mutant (which we do in new experiments, now Fig. S1). We also note that some of the interpretations of these data were made challenging by result variability.

To help address these issues we performed additional replicates for all of our explant experiments (contained within Figure 1, Fig. S1-S2, and Data S1), to provide greater power for our analyses. These new data provide a clearer view of this system that revise our interpretations from the prior version of this study. While treatment with indole alone does suppress the WT advantage over chemotactic mutants for both total colonization and cellular invasion, essentially all other treatments have a similar result with a timedependent increase in both colonization and invasion, dependent on chemotaxis and Tsr. A remaining unique feature of fecal treatment is an increase in the cellular invaded population of the cells at 3 h post-infection. As requested by Reviewer 3, we provide new experimental data showing that in competitions between WT and an invasion-deficient mutant (invA), with fecal material pretreatment, we see the WT has an advantage only for the gentamicin-treated qualifications, providing some support that our model selects for the invaded sub-population. Although we note that the invA still can invade through alternative mechanisms (as discussed in earlier work such as here: https://doi.org/10.1111/1574-6968.12614), so the relative amount of presumed cellular invasion is less than WT, and not zero, in our experiments (Fig. S1).

One point of confusion in the previous version of the text was the assay design for the explant experiments, which is important to understand in order to interpret the results. During the explant infection bacteria are not immersed in the effector treatment solution, rather the tissue is soaked in the effector solution beforehand and then exposed to a 300 µl buffer solution containing the bacteria. This means that the bacteria experience only the residue of that treatment at concentrations far lower. We have added clarity about this through revising Fig. 1 to include a conceptual diagram of the assay (Fig. 1C), and added a new supplementary Fig. S5 that summarizes the explant data in this same conceptual model. We provide detail on the method in the text in lines 115-137. In describing the results, and synthesizing them in the discussion, we now state:

Line 112: “This establishes a chemical gradient which we can use to quantify the degree to which different effector treatments are permissive of pathogen association with, and cellular invasion of, the intestinal mucosa (Fig. 1C).”

And, a new section in the discussion devoted to describing the explant infections:

Line: 366: “Our explant experiments can be thought of as testing whether a layer of effector solution is permissive to pathogen entry to the intestinal mucosa, and whether chemotaxis provides an advantage in transiting this chemical gradient to associate with, and invade, the tissue (Fig. 1C, Fig. S5).”

As mentioned above, we have honed the text to focus on the disparity between the effects of indole alone versus treatments with indole-rich feces to help clarify how these data advance our understanding of the indole taxis in directing pathogenesis. While our explant studies still confirm the role of factors other than L-Ser, indole, and Tsr in directing Salmonella infection and cellular invasion, we now include further analyses of other fecal effectors (described above) that provide some insights into how fecal effectors have some redundancy in their impact.

**Public Reviews:**

**Reviewer #1 (Public review):**
Summary:The study shows, perhaps surprisingly, that human fecal homogenates enhance the invasiveness of Salmonella typhimurium into cells of a swine colonic explant. This effect is only seen with chemotactic cells that express the chemoreceptor Tsr. However, two molecules sensed by Tsr that are present at significant concentrations in the fecal homogenates, the repellent indole and the attractant serine, do not, either by themselves or together at the concentrations in which they are present in the fecal homogenates, show this same effect. The authors then go on to study the conflicting repellent response to indole and attractant response to serine in a number of different in vitro assays.Strengths:The demonstration that homogenates of human feces enhance the invasiveness of chemotactic Salmonella Typhimurium in a colonic explant is unexpected and interesting. The authors then go on to document the conflicting responses to the repellent indole and the attractant serine, both sensed by the Tsr chemoreceptor, as a function of their relative concentration and the spatial distribution of gradients.

Thank you for your summary and acknowledgement of the strengths of this work. We hope the revised text and additional data we provide further improve your view of the study.

Weaknesses:The authors do not identify what is the critical compound or combination of compounds in the fecal homogenate that gives the reported response of increased invasiveness. They show it is not indole alone, serine alone, or both in combination that have this effect, although both are sensed by Tsr and both are present in the fecal homogenates. Some of the responses to conflicting stimuli by indole and serine in the in vitro experiments yield interesting results, but they do little to explain the initial interesting observation that fecal homogenates enhance invasiveness.

Thank you for noting these weaknesses. We have provided new data using a defined mixture of fecal effectors to further investigate the roles of L-Ser, indole, and other effectors present in feces that we did not initially study. We have refined our discussion of these results to hopefully improve the clarity of our conclusions. We show now both in explant studies (Fig. 1I) and chemotaxis responses to a defined fecal effector system (Fig. 5) that L-Ser is able to abolish both the suppression of indole-mediated WT advantage and also indole chemorepulsion, respectively. We also show the latter can be accomplished by other fecal chemoattractants (Fig. 5). This is in line with our earlier finding that Tsr, the sensor of indole and L-Ser, is an important mediator of fecal attraction but not the sole mediator.

As this reviewer points out, there are indeed other factors mediating invasion that we do not elucidate here, but we do note these possibilities in the text (lines: 125-127):

“This benefit may arise from a combination of factors, including sensing of host-emitted effectors, redox or energy taxis, and/or swimming behaviors that enhance infection [5,30,31,35].”

**Reviewer #2 (Public review):**
Summary:The manuscript presents experiments using an ex vivo colonic tissue assay, clearly showing that fecal material promotes Salmonella cell invasion into the tissue. It also shows that serine and indole can modulate the invasion, although their effects are much smaller. In addition, the authors characterized the direct chemotactic responses of these cells to serine and indole using a capillary assay, demonstrating repellent and attractant responses elicited by indole and serine, respectively, and that serine can dominate when both are present. These behaviors are generally consistent with those observed in *E. coli*, as well as with the observed effects on cell invasion.Strengths:The most compelling finding reported here is the strong influence of fecal material on cell invasion. Also, the local and time-resolved capillary assay provides a new perspective on the cell's responses.

Thank you for acknowledging these aspects of the study.

Weaknesses:The weakness is that indole and serine chemotaxis does not seem to control the fecal-mediated cell invasion and thus the underlying cause of this effect remains unclear.In addition, the fact that serine alone, which clearly acts as a strong attractant, did not affect cell invasion (compared to buffer) is somewhat puzzling. Additionally, wild-type cells showed nearly a tenfold advantage even without any ligand (in buffer), suggesting that factors other than chemotaxis might control cell invasion in this assay, particularly in the serine and indole conditions. These observations should probably be discussed.

Addressed above.

Final comment. As shown in reference 12, Tar mediates attractant responses to indole, which appear to be absent here (Figure 3J). Is it clear why? Could it be related to receptor expression?

Thank you for noting this. We now mention this in the discussion. In the course of this work, we encountered a number of apparent inconsistencies, or differences, between what we were observing with S. Typhimurium and what had been reported previously in studies of Tsr function in *E. coli*. We indeed noted that some studies had investigated a role of Tar for indole taxis (in *E. coli*), hence why we determined whether, and confirmed, that Tsr is required for indole taxis for S. Typhimurium (Fig. 6).

We do not know the reason for this apparent difference between the two bacteria, but we have previously shown with our same strain of S. Typhimurium IR715, under the same growth assay, and preparation protocol, that L-Asp is a strong chemoattractant for both WT and the tsr mutant (see Glenn et al. 2024, eLife, Fig. 5G: https://iiif.elifesciences.org/lax:93178%2Felife-93178-fig5-v1.tif/full/1500,/0/default.jpg).

This supports that this strain of Salmonella indeed has a functional Tar present and is expressed at a level sufficient for sensing L-Asp. So, if Tar generally mediates indole sensing we do not know why we would not see that in Salmonella. Hence, we do not see any role for Tar in indole chemorepulsion in our strain of study, which is different than reported for *E. coli*, but we cannot confirm the reason.

**Reviewer #3 (Public review):**
Summary:In this manuscript, Franco and colleagues describe careful analyses of Salmonella chemotactic behavior in the presence of conflicting environmental stimuli. By doing so, the authors describe that this human pathogen integrates signals from a chemoattractant and a chemorepellent into an intermediate "chemohalation" phenotype.Strengths:The study was clearly well-designed and well-executed. The methods used are appropriate and powerful. The manuscript is very well written and the analyses are sound. This is an interesting area of research and this work is a positive contribution to the field.

Thank you for your comments.

Weaknesses:Although the authors do a great job in discussing their data and the observed bacterial behavior through the lens of chemoattraction and chemorepulsion to serine and indole specifically, the manuscript lacks, to some extent, a deeper discussion on how other effectors may play a role in this phenomenon. Specifically, many other compounds in the mammalian gut are known to exhibit bioactivity against Salmonella. This includes compounds with antibacterial activity, chemoattractants, chemorepellers, and chemical cues that control the expression of invasion genes. Therefore, authors should be careful when making conclusions regarding the effect of these 2 compounds on invasive behavior.

Thank you for this comment, and we agree with your point. We hope we have revised the text and provided new data to address your concern. We have also chosen for clarity to keep our text close to our experimental data and so have refrained from speculating about some topics, even though you are absolutely correct about the immense complexity of these systems.

It is important that the word invasion is used in the manuscript only in its strictest sense, the ability displayed by Salmonella to enter non-phagocytic host cells. With that in mind, authors should discuss how other signals that feed into the control of Salmonella invasion can be at play here.

Thank you for your recommendation. We have revised the text to hopefully be clearer on our meaning of invasion in regard to Salmonella entering non-phagocytic host cells, essentially changing our usage to ‘cellular invasion’ throughout.

It is also a commonly-used phrase in reference to enteric infections and the colonization resistance conferred by the microbiome to refer to ‘invading pathogens’ (i.e. invasion in the sense of a new microbe colonizing the intestines), For instance, this recent review on Salmonella makes use of the term invading pathogen (https://www.nature.com/articles/s41579-021-00561-4). We acknowledge the confusion by this dual use of the term. We have mostly removed our statements using invasion in this context. We hope our language is clearer in this revised version.

**Recommendations for the authors:**

**Reviewer #1 (Recommendations for the authors):**
It was difficult to understand the true intent or importance of the study described in this manuscript. The first figure in the paper showed that a Salmonella Typhimurium strain lacking either CheY, and thus incapable of any chemotaxis, or the Tsr chemoreceptor, and thus incapable of sensing serine or indole, was modestly inferior to the wild-type version of that strain in invading the cells of a swine colonic explant. It then showed that, in the presence of a human fecal homogenate, the wild-type strain had a much greater advantage in invading the colonic cells. Thus, the presence of the fecal homogenate significantly increased invasiveness in a way that depends on chemotaxis and the Tsr chemoreceptor.As human feces were determined to contain 882 micromolar indole and 338 micromolar serine, the effects of those concentrations of either indole or serine alone or in combination were tested. The somewhat surprising finding was that neither indole nor serine alone nor in combination changed the result from the experiment done with just buffer in the colonic explant.The clear conclusion of this initial study is that both chemotaxis in general and chemotaxis mediated by Tsr improve the invasiveness of S. Typhimurium. They provide a much bigger advantage in the presence of human feces. However, two molecules present in the feces that are sensed by Tsr, serine, and indole, seem to have no effect on invasiveness either alone or in combination.At this point, the parsimonious interpretation is that there is something else in human feces that is responsible for the increased invasiveness, and the authors acknowledge this possibility. However, they do not take what appears to be the obvious approach: to look for additional factors in human feces that might be responsible, either by themselves or in combination with indole and/or serine, for the increased invasiveness. Instead, they carry out a detailed examination of the counteracting effects of indole as a repellent and of serine as an attractant as a function of their relative concentrations and their spatial distributions.

Thank you for your comments. In our revised version, we have undertaken some additional studies of other fecal effectors that help better understand the relationship between L-Ser and indole, but also the roles of other chemoattractants (glucose, galactose, ribose, L-Asp) in mediating fecal attraction (Fig. 5). We agree with the reviewer and conclude that fecal attraction and the cell invasion phenotype mediated by fecal treatment are influenced by factors other than only Tsr, indole, and L-Ser. Our new data do show that L-Ser is sufficient to block both the invasion suppression effects of indole (negating the WT advantage) and also indole chemorepulsion, therefore making our detailed examination of the counteracting effects more relevant for understanding this system.

What they find is what other studies have shown, primarily with S. Typhimurium's relative, the gamma-proteobacterium *Escherichia coli*.At high indole and low serine concentrations, the repulsion by indole wins out. At low indole and high serine concentrations, attraction by serine wins out. What is perhaps novel is what happens at an intermediate ratio of concentrations. Repulsion by indole dominates at short distances from the source, so there is a zone of clearing. At longer distances, attraction by serine dominates, so there is an accumulation of cells in a "halo" around the zone of clearing. Thus, assuming that serine and indole diffuse equally, the repulsive effect of indole dominates until its concentration falls below some critical level at which the concentration of serine is still high enough to exert an attractive effect.They go on to show, using ITC, that serine binds to the periplasmic ligand-binding domain (LBD) of Tsr, something that has been studied extensively with very similar *E. coli* Tsr.They also show that indole does not bind to the Tsr LBD, which also is known for *E. coli* Tsr.This would be newsworthy only if the results were different for S. Typhimurium than for *E. coli*. As it is, it is merely confirmatory of something that was already known about Tsr of enteric bacteria.An idea that the authors introduce, if I understand it correctly, is that a repellent response to something in feces, perhaps indole, drives S. Typhimurium chemotactically competent cells out of the colonic lumen and promotes invasion of the bacteria into the cells of the colonic lining. If the feces contain both an attractant and a repellent, bacteria might be attracted by the feces to the lining of the intestine and then enter the colonic cells to escape a repellent, perhaps indole. That is an interesting proposition.In summary, I think that the initial experimental approach is fine. I do not understand the failure to follow up on the effect of the fecal homogenates in promoting invasion by chemotactic bacteria possessing Tsr. It seems there must be something else in the homogenates that is sensed by Tsr. Other amino acids and related compounds are also sensed by Tsr. Perhaps it is energy or oxygen taxis, which is partially mediated by Tsr, as the authors acknowledge.Much of the work reported here is quasi-repetitive with work done with *E. coli* Tsr. Minimally, previous work on *E. coli* Tsr should be explained more thoroughly rather than dealt with only as a citation.

Thank you for your comments.

We would like to confirm our agreement that *E. coli* and *S. enterica* indeed possess similarities. They are Gammaproteobacteria and inhabit/infect the gut. But also we note they diverged evolutionarily during the Jurassic period (ca. 140 million years ago, see: PMC94677). In the context of colonizing humans, the former is a pathobiont, indoleproducer, and a native member of the microbiome, whereas the latter is a frank pathogen and does not produce indole. Hence, there are many reasons to believe one is not an approximate of the other, especially when it comes to causing disease.

We agree that much of what is known about indole taxis has come from excellent studies in well-behaved laboratory strains of *E. coli*, a powerful model. We believe that expanding this work to include clinically relevant pathogens is important for understanding its role in human disease. In this study, we contribute to that broader understanding by providing new mechanistic insights into Tsr-mediated indole taxis in S. Typhimurium, along with data demonstrating fecal attraction in other enteric pathogens and pathobionts. These findings help define a more general role for Tsr in enteric colonization and disease. While some of our results indeed confirm and extend prior findings, we respectfully believe that such confirmation in relevant pathogenic strains adds value to the field.

Regarding our ITC studies, to our knowledge no other study has investigated, using ITC whether indole does or does not bind the LBD (which we show it does not), nor investigated whether it interferes with L-Ser sensing (which we show it does not). Hence, these are not duplicate findings, although we do acknowledge this leaves the mechanism of indolesensing undiscovered. If we are incorrect in this regard, please provide us a citation and we will be happy to include it and revise our comments.

We now clarify in the text on lines 378-381: “While these leave the molecular mechanism of indole-sensing unresolved, it does eliminate two possibilities that have not, to our knowledge, been tested previously. Overall, our data add support to the hypothesis that a non-canonical sensing mechanism is employed by Tsr to respond to indole [8,18,69].”

Lastly, as noted by the reviewer, and which we mention in the text, essentially all prior studies on indole taxis were conducted in *E. coli*, and this is not what is new and novel about the work we present, which is focused on S. Typhimurium and testing the prediction that fecal indole protects against pathogen invasion. We have added in a few additional points of comparisons between our results and prior studies. While we appreciate that much understanding has come from *E. coli* as a model for indole taxis, we feel discussing prior work in extensive detail would be more suitable for a review and would occlude our new findings about Salmonella, and other enterics.

In an earlier version of the manuscript, we included more background on *E. coli* indole taxis. However, we found that the historical literature in this area was somewhat inconsistent, with different assays using varying time points and indole concentrations, often leading to results that were difficult to reconcile. Providing sufficient context to explain these discrepancies required considerable space and, ultimately, detracted from the focus of our current study. Hence, we have only brought in comparisons with *E. coli* where most relevant to the present work. Also, we provide new data that *E. coli* also exhibits fecal attraction, and so there is reason to believe the mechanisms we study here are also relevant to that system.

Some minor points(1) Hyphens are not needed with constructs like "naturally occurring" or "commonly used".

Thank you. Revisions made throughout.

(2) The word "frank" as in "frank pathogen" seems odd. It seems "potent" would be better.

Thank you for this comment. Per your recommendation, we have removed this term.

The term ‘frank pathogen’ is standard usage in the field of bacterial pathogenesis in reference to a microbe that always causes disease in its host (in this case humans) and causes disease in otherwise healthy hosts (example: https://www.sciencedirect.com/science/article/pii/S1369527420300345). We actually used this specific term to distinguish an aspect of novelty of our study because *E. coli* can, sometimes, be a pathogen (i.e. a pathobiont) and of course *E. coli* indole taxis has been previously studied. Ours is the first study of indole taxis in a frank pathogen.

(3) It is unnecessary to coin a new word, chemohalation, to describe a phenomenon that is a simple consequence of repulsion by higher concentrations of a repellent and attraction by lower concentrations of attractant to generate a halo pattern of cell distribution.

Thank you for your opinion on this. We have softened our statements on this point, and in the newly revised version of the text less space is devoted to this idea. We now state in line 304-307:

“There exists no consensus descriptor for taxis of this nature, and so we suggest expanding the lexicon with the term “chemohalation,” in reference to the halo formed by the cell population, and which is congruent with the commonly-used terms chemoattraction and chemorepulsion.”

We appreciate the reviewer’s perspective and agree that the behavior we describe can be viewed as the result of competing attractant and repellent cues. However, we find that the traditional framework of “chemoattraction” and “chemorepulsion” is often insufficient to describe the spatial positioning behaviors we observe in our system. In our experience presenting and discussing this work, especially with audiences outside the chemotaxis field, it has been challenging to convey these dynamics clearly using only those two terms.

For this reason, we introduced the term chemohalation to describe this more nuanced behavior, which appears to reflect a balance of signals rather than a simple unidirectional response. More bacteria enter the field of view, but they are clearly positioned differently than regular ‘chemoattraction.’ We also note that Reviewers 2 and 3 did not raise concerns about the term, and after careful consideration, we have opted to retain it in the revised manuscript.

**Reviewer #2 (Recommendations for the authors):**
Lines 143-156 seem somewhat overcomplicated and may be confusing. For example: in line 143: "However, when colonic tissue was treated with purified indole at the same concentration, the competitive advantage of WT over the chemotactic mutants was abolished compared to fecaltreated tissue...". But indole was tested alone, so it did not abolish the response; rather the absence of fecal material did.

We appreciate your point. We have made revisions throughout to help improve the clarity of how we discuss the explant infection data and provide new visuals to help explain the experiment and data (Fig. 1C, Fig. S5).

**Reviewer #3 (Recommendations for the authors):**
(1) Line 46 - Are references 9-11 really about topography?

Thank you. You are correct. Revised and eliminated this statement.

(2) Lines 87-89 - It seems to me that a bit more information on this would be helpful to the reader.

In our revision of the text, to make it more centered on our primary findings of the differences between indole taxis when indole is the sole effector versus amidst other effectors, we have removed this section.

(3) Line 112 - When mentioning the infection of the cecum and colon, authors should specify that this is in mice.

Thank you for this comment. In our revised version we provide references both for animal model infections and work in human patients (ex: https://www.sciencedirect.com/science/article/abs/pii/S0140673676921000)

We have revised our statement to be (Line 99-100: “Salmonella Typhimurium preferentially invades tissue of the distal ileum but also infects the cecum and colon in humans and animal models [42–46]).”

(4) Lines 122-123 - Authors state that "This experimental setup simulates a biological gradient in which the effector concentration is initially highest near the tissue and diffuses outward into the buffer solution.". Was this experimentally demonstrated? If not, authors should tone this down.

We have removed this comment and instead present a conceptual diagram illustrating this idea (Fig. 1C). Also, addressed by above.

(5) When looking at the results in Figure 1, I wonder what the results of this experiment would be if the authors tested an invasion mutant of Salmonella. In a strain that is able to perform chemotaxis (attraction and repulsion) but unable to actively invade, would there be a phenotype here? Is it possible that the fecal material affects cellular uptake of Salmonella, independently of active invasion? I don't think the authors necessarily need to perform this experiment, but I think it could be informative and this possibility should at least be discussed.

Thank you for your comments and suggestions. We have included new data of an explant co-infection experiment with WT and an invasion-deficient mutant invA (Fig. S1). Under these conditions, WT exhibits an advantage in the gentamicin-treated homogenate, but not the untreated homogenate, suggestive of an advantage in cellular invasion.

However, we did not repeat all experiments with this genetic background. We felt that would be outside the scope of this work, and would probably require dual chemotaxis/invA deletions to assess the impact of each, which also could be difficult to interpret. The hypothesis mentioned by the Reviewer is possible, but we were not able to devise a way to test this idea, as it seems we would need to deactivate all other mechanisms of Salmonella invasion.

(6) Lines 137-140 - Because this is a competition experiment and results are plotted as CI, the reader can't readily assess the impact of human feces on invasion by WT Salmonella.

Thank you for pointing this out. We want to mention that the data are plotted as CI in the main text, but the supplemental contains the disaggregated CFU data (Fig. S1-2) and the numerical values (Data S1).

Please include the magnitude of induction in this sentence, compared to the buffer control.

The text of this section has been changed to account for new data.

Additionally, although unlikely, the presence of the chemotaxis mutants in the same infection may be a confounding factor. In order to irrefutably ascertain that feces induces invasion, I suggest authors perform this experiment with the wildtype strain (and mutant) alone in different conditions.

Thank you for this suggestion, although after careful consideration we have decided not to repeat these explant studies with monoinfections. Coinfections are a common tool in *Salmonella* pathogenesis studies, including prior chemotaxis studies which our work builds upon (ex: https://pmc.ncbi.nlm.nih.gov/articles/PMC3630101/). The explant experiments, even controlling as many aspects as we did, still show lots of variability and one way to mitigate this is through competition experiments so that each strain experiences the same environment.

We agree that a cost of this approach is that one strain may affect the other, or may alter the environment in a way that impacts the other. Thus, the resulting data must also be understood through this lens. We have revised the text to stay closer to the competitive advantage phenotype.

(7) Line 150 - Authors state that bacterial loads are similar. However, authors should perform and report statistical analyses of these comparisons, at least in the supplementary data.

We have removed this statement as requested. We do note, however, that the mean CFU values across treatments at identical time points appear qualitatively similar, which is an observation that does not require statistical testing.

(8) Lines 154-154 - This seems incorrect, as the effect observed with the mixture of indole and serine is very similar to the addition of serine alone. Therefore, there was no "neutralization" of their individual effects.

We have revised this statement.

(9) Line 159-161 - I strongly suggest authors reword this sentence. I don't think this is the best way to describe these results. The stronger phenotype observed was with the fecal material. Therefore, it is the indole (alone) condition that does not "elicit a response". Focusing on indole too much here ignores everything else that is present in feces and also the fact that there was a drastic phenotype when feces were used.

Thank you for your opinion on this. We believe this is one of the ways in which our earlier draft was unclear. It was actually a primary motivation of this work to test whether there were differences in pathogen infection, mediated by chemotaxis, in the presence of indole as a singular effector or in its near-native context in fecal material, and our revised text centers our study around this question. We believe this distinction is important for the reasons mentioned earlier.

Relative to buffer treatment, indole changes the behavior of the system, eliminating the WT advantage, and this is the effect we refer to. We have made many revisions to the text of these sections and hope it better conveys this idea. We expect we may still have differences regarding the interpretation of these results, but regardless, thank you for your suggestions and we have tried to implement them to improve the clarity of the text.

(10) Line 162 - Again, I disagree with this. Indole does not have an effect to be cancelled out by serine.

Addressed above, and this text has been changed. Also, we provide new chemotaxis data that at fecal-relevant concentrations of indole and L-Ser, indole chemorepulsion is overridden (Fig. 5).

(11) Lines 166-168 - Again, this is a skewed analysis. Indole and serine could not possibly provide an "additive effect" since they do not provide an effect alone. There is nothing to be added.

This text has been deleted.

(12) Lines 168-170 - Most of the citations provided to this sentence are inadequate. Our group has previously shown that the mammalian gut harbors thousands of small molecules (Antunes LC et al. Antimicrob Agents Chemother 2011). You obviously do not have to cite our work, but there is significant literature out there about the complexity of the gut metabolome.

Thank you for this comment. We have revised this particular text, but do make mention of potential other effectors driving these effects, which was also requested by the other reviewers.

Your work and others indeed support there being thousands of molecules in the gut, but our work centers on chemotaxis, and bacteria have a small number of chemoreceptors and only sense a very tiny fraction of these molecules as effectors. Since the impacts of infection of the explants depends on chemotaxis, we keep our comments restricted to those, but agree that there are likely many interactions involved, such as those impacting gene expression.

Please note our more detailed description of the explant infection assay (and shown in Fig. 1C) that may change your view on the significance of non-chemotaxis effects. The bacteria only experience the effectors at low concentration, not the high concentration that is used to soak and prepare the tissue prior to infection.

(13) Figure 2 - The letter 'B' from panel B is missing.

Thank you very much for bringing this oversite to our attention. We have fixed this.

(14) Legend of Figure 3 - Panel J is missing a proper description. Figure legends need improvement in general, to increase clarity.

Thank you for noting this. This is now Fig. 6E. We have provided an additional description of what this panel shows. We have edited the legend text to read: “E. Shows a quantification of the relative number of cells in the field of view over time following treatment with 5 mM indole for a competition experiment with WT and tsr (representative image shown in F).”

We also have made other edits to figure legends to improve their clarity and add additional experimental details and context. By breaking up larger figures into smaller figures, we also hope to have improved the clarity of our data presentation.

(15) Lines 264-265 - Maybe I am missing something, but I do not see the ITC data for serine alone.

We have clarified in the text that this was measured in our previous study https://elifesciences.org/articles/93178. The present study is a ‘Research Advance’ article format, and so builds on our prior observation.

We have revised the text to read: “To address these possibilities, we performed ITC of 50 μM Tsr LBD with L-Ser in the presence of 500 μM indole and observed a robust exothermic binding curve and KD of 5 µM, identical to the binding of L-Ser alone, which we reported previously (Fig. 6H) [36].”

(16) Lines 296-297 - What is the effect of these combinations of treatments on bacterial cells? I commend the authors for performing the careful growth assays, but I wonder if bacterial lysis could be a factor here. I am not doubting the effect of chemotaxis, but I am wondering if toxic effects could be a confounding factor. For instance, could it be that the "avoidance" close to the compound source and subsequent formation of a halo suggest bacterial death and lysis? I suggest the authors perform a very simple experiment, where bacteria are exposed to the compounds at various concentrations and combinations, and cells are observed over time to ensure that no bacterial lysis occurs.

Thank you for mentioning this possibility. If we understand correctly, the Reviewer is asking if the chemohalation effect we report could be from the bacteria lysing near the source. Our data actually argue against this possibility through a few lines of evidence.

First, if this were the case in experiments with the cheY mutant, we would also see an effect near the source. But actually, in experiments with either the cheY mutant or the tsr mutant, neither of which can sense indole, the bacteria just ignore the stimulus and show an even distribution (see current Fig. 6F).

Second, our calculations suggest that in the chemotaxis assay (CIRA), the bacteria only experience rather low local concentration of indole, mostly I the nM concentration range, because as soon as the effector treatment is injected into the greater volume, it is immediately diluted. This means the local concentration is far below what we see inhibits growth of the cells in the long run and may not be toxic (Fig. 7, Fig. S3).

Lastly, in the representative video presented we can observe individual cells approach and exit the treatment (Movie 11). Due to the above we have not performed additional experiments to test for lysis.

(17) Lines 310-311 - Isn't this the opposite of the model you propose in Figure 5? The higher the concentration of indole in the lumen the more likely Salmonella is to swim away from it and towards the epithelium, favoring invasion, no?

We appreciate the opportunity to clarify this point and apologize for any confusion caused. In response, we have revised the text to place less emphasis on chemohalation, and the specific statement and model in question have now been removed. Instead, we provide a summary of our explant data in light of the other analyses in the study (Fig. S5).

What we meant here was in relation to the microscopic level, not whether or not a host/intestine is colonized. To put it another way, we think our data supports that the pathogen colonizes and infects the host regardless of indole presence, but it uses indole as a means to prioritize which tissues are optimal for colonization at the microscopic level. The prediction made by others was that bacteria swim away from indole source and therefor this could prevent or inhibit pathogen colonization of the intestines, which our data does not support.

(18) Lines 325-326 - Maybe, but feces also contain several compounds with antibacterial activity, as well as other compounds that could elicit chemorepulsion. This should be stated and discussed.

We have removed this statement since we did not explicitly test the growth of the bacteria with fecal treatments. We have refrained from speculating further in the text since we do not have direct knowledge of how that relationship with differing effectors could play out.

We agree with the reviewer that the growth assays are reductionist and give insight only into the two effectors studied. We provide evidence from several different types of enterics that they all exhibit fecal attraction, and it seems unlikely the bacteria would be attracted to something deleterious, but we have not confirmed.

(19) Lines 371-374 - How preserved (or not) is the mucus layer in this model? The presence of an inhibitory molecule in the lumen does not necessarily mean that it will protect against invasion. It is possible that by sensing indole in the lumen Salmonella preferentially swims towards the epithelium, thus resulting in enhanced evasion.

The text in question has been removed. However, we acknowledge the reviewer’s point, and that these explant tissues do not fully model an in vivo intestinal environment. Other than a gentle washing with PBS to remove debris prior to the experiment the tissue is not otherwise manipulated, and feasibly the mucus layer is similar to its in vivo state.

In mentioning this hypothesis about indole, which our data do not support, we were echoing a prediction from the field, proposed in the studies we cite. We agree with the reviewer that there were other potential outcomes of indole impacting chemotaxis and invasion, and indeed our data supports that.

(20) Lines 394-395 - The authors need to remember that the ability to invade the intestinal epithelium is not only a product of chemoattraction and repulsion forces. Several compounds in the gut are used by Salmonella as cues to alter invasion gene expression. See PMID: 25073640, 28754707, 31847278, and many others.

Thank for you for this point, and we now include these citations. We have revised the text in question, stating:

“In addition to the factors we have investigated, it is already well-established in the literature that the vast metabolome in the gut contains a complex repertoire of chemicals that modulate Salmonella cellular invasion, virulence, growth, and pathogenicity [79–81].”

Our intent is not to diminish the role of other intestinal chemicals but rather to put our new findings into the context of bacterial pathogenesis. We do provide evidence that specific chemoeffectors present in fecal material alter where bacteria localize through chemotaxis, which is one method of control over colonization.

(21) Line 408 - I think it could be hard to observe this using your experimental approach.Because you need to observe individual cells, the number of cells you observe is relatively small. If, in a bet-hedging strategy, the proportion of cells that were chemoattracted to indole was relatively low you likely would not be able to distinguish it from an occasional distribution close to the repellent source. You may or may not want to discuss this.

Thank you for this observation. It is indeed challenging to both observe large scale population behaviors and also the behaviors of individual cells in the same experiment. Our ability to make this distinction is similar to the approach used in the study we cite, so that is our comparison.

But, if there was a subpopulation that was attracted we would predict a ‘bull’s-eye’ population structure, with some cells attracted and other avoiding the source, which we do not see - we see the halo. So, we find no evidence of the bet-hedging response seen in a different study using *E. coli* and using different time scales than we have.

(22) Lines 410-411 - What could the other attractants be? Would it be possible/desirable to speculate on this?

We have changed the text here, but we present new data that examines some of these other attractants (Fig. 5).

(23) Line 431 - What exactly do you mean by "running phenotype"? Please, provide a brief explanation.

We have removed this text, but a running phenotype means the swimming bacteria rarely make direction changes (i.e. tumbles), which has been associated with promoting contact with the epithelium, described in the references we cite. Hence, this type of swimming behavior could contribute to the effects we observe in the explant studies, potentially explaining some of the Tsr-mediated advantage that was not dependent on L-Ser/indole.

(24) Line 441 - Other work has shown that feces contain inhibitors of invasion gene expression. The authors should integrate this knowledge into their model. In fact, indole has been shown to repress host cell invasion by Salmonella, so it is important that authors understand and discuss the fact that the impact of indole is multifaceted and not only a reflection of its action as a chemorepellent. PMID: 29342189, 22632036.

We agree with the reviewer about this point, and mention this in the text (lines 55-57): “Indole is amphipathic and can transit bacterial membranes to regulate biofilm formation and motility, suppress virulence programs, and exert bacteriostatic and bactericidal effects at high concentrations [16–18,20–22].”

We have added in the references suggested.

What we test here is the specific hypothesis made by others in the field about indole chemorepulsion serving to dissuade pathogens from colonizing.

For instance, the statement from: https://journals.plos.org/plosone/article?id=10.1371/journal.pone.0190613

“Since indole is also a chemorepellent for EHEC [23], it is intriguing to speculate that in addition to attenuating Salmonella virulence, indole also attenuates the recruitment and directed migration of Salmonella to its infection niche in the GI tract.”

And from: https://doi.org/10.1073/pnas.1916974117

“We propose that indole spatially segregates cells based on their state of adaptation to repel invaders while recruiting beneficial resident bacteria to growing microbial communities within the GI tract.”

And

“Thus, foreign ingested bacteria, including invading pathogens such as *E. coli* O157:H7 and *S. enterica*, are likely to be prevented by indole from gaining a foothold in the mucosa.”

As shown by others, indole certainly does have many roles in controlling pathogenesis, and there are other chemicals we do not investigate that control invasion and bacterial growth, but we keep our statements here restricted to chemotaxis since that is what are experiments and data show.

(25) Line 472 - "until fully motile". How long did this take, how variable was it, and how was it determined?

Thank you for asking for this clarification. We have added that the time was between 1-2 h, and confirmed visually. Our methods are similar to those described in earlier chemotaxis studies (ex: 10.1128/jb.182.15.4337-4342.2000).

(26) Line 487 - I worry that the fact fecal samples were obtained commercially means that compound stability/degradation may be a factor to consider here. How long had the sample been in storage? Is this information available?

Thank you for this question. We agree that the fecal sample we used serves as a model system and we cannot rule out that handling by the supplier could potentially alter its contents in some way that would impact bacterial chemosensing. However, we note that the measurements of L-Ser and indole we obtained are in the appropriate range for what other studies have shown.

The fecal sample used for all work in the study were from a single healthy human donor, obtained from Lee Biosolutions (https://www.leebio.com/product/395/fecal-stool-samplehuman-donor-991-18). The supplier did not state the explicit date of collection, nor indicated any specific handline or storage methods that would obviously degrade its native metabolites, but we cannot rule that out. In our hands, the fecal sample was collected and kept frozen at -20 C. For research purposes, portions were extracted and thawed as needed, maintaining the frozen state of the original sample to limit degradation from freeze-thaws.